# Seeking Commonality, Preserving Specificity: A Spectral-Aware Hierarchical Framework for Cross-City Road Representation Learning

**Jingtian Ma** [1]  **Jingyuan Wang** [1 2 3]  **Leong Hou U** [4]

## Abstract

Learning unified road representations across diverse cities is a pivotal challenge in urban computing. However, existing approaches predominantly focus on single-city modeling, failing to handle the distribution shifts caused by heterogeneous urban layouts. We identify *spectral misalignment*, manifested as the significant divergence of spectral distributions across different cities, as the primary barrier preventing standard Graph Neural Networks from capturing universal patterns. To bridge this gap, we propose **CoSpec**, a framework that disentangles road networks into shareable low-frequency commonalities and city-specific high-frequency specificities. CoSpec employs a hierarchical dual-path architecture where the low-frequency path aligns global functional semantics via adaptive prototypes, while the high-frequency path modulates local geometric residuals to fit specific urban textures. Theoretical analysis shows CoSpec bounds the Wasserstein distance between city distributions, and extensive experiments demonstrate its superior generalization over state-of-the-art baselines.

## 1. Introduction

Road networks form the backbone of modern cities and support a wide range of urban computing applications (Wang et al., 2021a; Chen et al., 2025; Ma & Wang, 2026), including traffic flow forecasting (Ji et al., 2020; Ali et al., 2025; Jiang et al., 2023a), route planning (Wu et al., 2019; Wang et al., 2021b; Yu et al., 2025), and POI recommendation (Zhang et al., 2025; Cheng et al., 2025), all of which

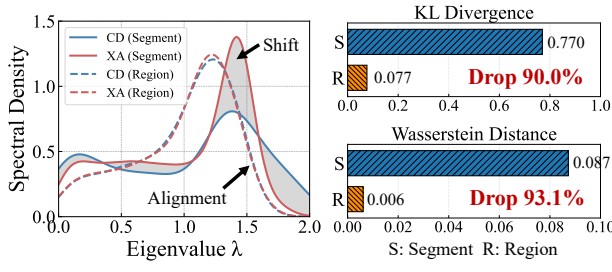

Figure 1. Spectral analysis of cross-city road networks.

rely on learning effective road representations. While existing studies have made substantial progress in *single-city* road representation learning, they typically model each city in isolation. In contrast, *Unified Cross-City Road Representation Learning*, which means learning a single model that generalizes across heterogeneous urban layouts, remains largely unexplored. Solving this problem is crucial for scalable urban intelligence, as it enables transferable structural patterns and shared functional semantics across cities.

However, achieving unified cross-city learning is fundamentally challenging due to severe *distribution shifts* induced by urban heterogeneity. Most existing road representation learning methods are originally designed for *single-city* modeling and fall into two categories, both of which struggle under such cross-city shifts. *Spatial-based GNNs* (Pei et al., 2020; Wu et al., 2020) emphasize local neighborhood aggregation and tend to overfit to city-specific geometric details, failing to capture macro-level functional isomorphisms shared across cities. In contrast, *spectral-based GNNs* (Kipf & Welling, 2017; He et al., 2021) aim to model global structures via Laplacian-based filters, but suffer from *spectral misalignment*: the Laplacian eigenbasis is strictly graph-dependent, rendering high-frequency components inherently non-transferable across cities. As a result, naively applying a single model across multiple cities induces destructive interference and leads to poor cross-city generalization.

A key empirical finding of this work emerges from the spectral analysis of road networks across different cities, as illustrated in Figure 1. At the *segment* level, spectral distributions exhibit pronounced cross-city divergence, reflecting substantial mismatch in microscopic topology. By contrast, when road segments are aggregated into *functional*

[1]School of Computer Science and Engineering, Beihang University, Beijing, China [2]School of Economics and Management, Beihang University, Beijing, China [3]MIIT Key Laboratory of Data and Decision Intelligence, Beihang University, Beijing, China [4]University of Macau, Macau SAR, China. Correspondence to: Jingyuan Wang <jywang@buaa.edu.cn>.

*Proceedings of the 43rd International Conference on Machine Learning*, Seoul, South Korea. PMLR 306, 2026. Copyright 2026 by the author(s).

*regions*, the spectral distributions across cities become well aligned over a broad range of eigenvalues. Figure 1(b) further shows that such region-level abstraction reduces cross-city discrepancy by over 90% under both KL divergence and Wasserstein distance. These results indicate that cross-city discrepancies are pronounced at the fine-grained segment level, whereas macro-scale region-level organization is substantially more consistent across cities. Viewed through the lens of hierarchical frequency decomposition (Ma et al., 2026), this suggests that region-level abstractions primarily capture transferable low-frequency functional patterns, while segment-level details are more associated with city-specific, high-frequency geometric variations.

Building on this insight, we propose **CoSpec** (**Co**mmonality-**Spec**ificity Disentanglement), a framework grounded in the principle of "*seeking commonality while preserving specificity*". CoSpec is built around two coupled designs. First, we construct a hierarchical abstraction that aggregates segments into functional regions and perform *region-level manifold alignment* by aligning cities in a shared region prototype space. Second, to preserve city-specific geometry without corrupting the shared manifold, we introduce a dual-path architecture that explicitly separates frequency components: a low-frequency path operates on region representations and enforces prototype-consistent alignment, while a high-frequency path models segment-level residuals and applies city-adaptive modulation to accommodate domain-specific variations. Our theory shows that region-level abstraction provably contracts the cross-city distribution discrepancy, and that the proposed disentanglement mechanism isolates non-transferable components, thereby enabling a single unified model to generalize across heterogeneous urban layouts.

Our contributions are fourfold: (1) We highlight the key insight that *spectral misalignment* is a fundamental barrier to cross-city road representation learning, and formulate unified modeling via disentanglement of low-frequency commonalities and high-frequency specificities. (2) We propose CoSpec, a spectral-aware hierarchical framework that integrates region-level alignment with dual-frequency modeling, enabling transferable yet city-adaptive road representations within a single model. (3) We provide theoretical analysis showing that region-level abstraction contracts the Wasserstein distance between city distributions. (4) We empirically demonstrate consistent improvements over state-of-the-art methods across multiple cities and downstream tasks.

## 2. Related Works

**Road Network Modeling.** Road network modeling provides the representation foundation for various downstream tasks in intelligent transportation systems (Ji et al., 2026; Ma et al., 2025; Jiang et al., 2023b; Ji et al., 2022; Yang et al., 2025). Early road network representation methods rely on shallow embeddings such as DeepWalk and Node2Vec (Per-

ozzi et al., 2014; Grover & Leskovec, 2016), which capture structural proximity but ignore attributes and functional semantics. Spatial GNNs (Wang et al., 2025; Han et al., 2025) incorporate node features through localized message passing, yet their limited receptive fields often cause over-smoothing and hinder long-range dependency modeling. Hierarchical frameworks (Ying et al., 2018; Wu et al., 2020) address this issue via graph coarsening across multiple scales. Recently, HiFiNet (Ma et al., 2026) introduces frequency decomposition to jointly model global trends and local variations. However, it is tailored to single-city settings and lacks mechanisms to enforce cross-city low-frequency consistency or preserve city-specific high-frequency characteristics, leading to spectral misalignment under transfer.

**Cross-City Learning.** Cross-city learning transfers knowledge from data-rich cities to alleviate data scarcity. Grid-based methods (Wang et al., 2019a; Fang et al., 2022) align regions using traffic statistics but discretize road networks into Euclidean grids, causing structural fragmentation. More advanced approaches employ meta-learning (Yao et al., 2019; Pan et al., 2019) or adversarial domain adaptation (Jin et al., 2022; Xu et al., 2024) to align feature distributions across cities. However, these methods operate primarily in the spatial domain and enforce indiscriminate alignment of all features, including city-unique high-frequency components. Such over-alignment erases local specificity and often results in negative transfer. Our CoSpec framework mitigates this issue via explicit spectral disentanglement.

**Graph Neural Networks.** GNNs can be broadly categorized into spatial and spectral approaches. Spatial GNNs (Velickovic et al., 2017; Hamilton et al., 2017) and recent Graph Transformers (Dwivedi & Bresson, 2021; Wu et al., 2022) enhance global modeling through attention, but incur high complexity and rely on positional encodings that generalize poorly across heterogeneous graphs. Spectral GNNs (Defferrard et al., 2016; Bo et al., 2021) capture global structure via Laplacian filtering, yet their filters are tightly coupled to graph-specific eigenbases, limiting cross-graph transfer. Our work bridges these limitations by learning shared spectral commonality while preserving graph-specific spectral specificity within a hierarchical framework.

## 3. Preliminaries

In this section, we introduce the basic concepts and notations used throughout the paper. We first formalize road networks with graph signals, then define their hierarchical abstraction, and finally present the unified cross-city representation learning problem.

### 3.1. Road Network with Graph Signals

We model a road network at the microscopic level as a directed graph endowed with node-wise signals, which serves

as the foundation for subsequent spectral analysis.

**Definition 3.1** (Road Graph Signal). A road network is represented as a directed graph $\mathcal{G}_S = (\mathcal{V}_S, \mathbf{A}_S, \mathbf{X}_S)$, where $\mathcal{V}_S$ denotes the set of $N_S$ road segments, $\mathbf{A}_S \in \{0,1\}^{N_S \times N_S}$ is the adjacency matrix encoding physical connectivity, and $\mathbf{X}_S \in \mathbb{R}^{N_S \times d_{in}}$ is the graph signal matrix. Each row $\mathbf{x}_i$ corresponds to the attribute vector of segment $v_i$.

**Spectral view.** From the perspective of graph signal processing (GSP), $\mathbf{X}_S$ can be interpreted as a superposition of spectral components defined over the graph Laplacian of $\mathcal{G}_S$. Low-frequency components typically capture smooth, large-scale functional patterns, while high-frequency components reflect localized geometric irregularities. This spectral characterization provides a principled lens for analyzing structural commonalities and variations across different cities.

### 3.2. Hierarchical Abstraction of Road Networks

Urban road networks exhibit inherent multi-scale organization. To explicitly model this property, we introduce a hierarchical abstraction that aggregates fine-grained road segments into coarse-grained functional regions.

**Definition 3.2** (Region Graph). A region graph is defined as $\mathcal{G}_R = (\mathcal{V}_R, \mathbf{A}_R, \mathbf{X}_R)$, where $\mathcal{V}_R = \{r_1, \ldots, r_{N_R}\}$ is the set of $N_R$ functional regions, $\mathbf{A}_R \in \mathbb{R}^{N_R \times N_R}$ encodes macroscopic connectivity between regions, and $\mathbf{X}_R \in \mathbb{R}^{N_R \times d}$ denotes region-level representations that summarize the aggregated semantics of each region.

**Definition 3.3** (Hierarchical Road Network). A hierarchical road network is modeled as a tuple $\mathcal{H} = (\mathcal{G}_S, \mathcal{G}_R, \mathbf{A}_{SR})$, where $\mathcal{G}_S$ and $\mathcal{G}_R$ denote the segment-level and region-level graphs, respectively, and $\mathbf{A}_{SR} \in \mathbb{R}^{N_S \times N_R}$ is a cross-scale assignment matrix. Each entry $\mathbf{A}_{SR}[i,j]$ represents the affiliation strength between segment $s_i$ and region $r_j$.

### 3.3. Problem Formulation

We consider the problem of learning unified road representations across heterogeneous cities.

**Definition 3.4** (Unified Cross-City Road Representation Learning). Let $\mathcal{C}$ denote a set of cities, where each city $c \in \mathcal{C}$ is represented by a hierarchical road network $\mathcal{H}^{(c)}$. The goal is to learn a unified, city-agnostic encoder $f_\theta : \mathcal{H}^{(c)} \to \mathbf{Z}_S^{(c)}$ that maps each city to segment-level representations $\mathbf{Z}_S^{(c)} \in \mathbb{R}^{N_S^{(c)} \times d}$, while capturing both city-invariant structural commonalities and city-specific characteristics.

## 4. Methodology

In this section, we present **CoSpec** (*Commonality-Specificity Disentanglement Network*). Instead of viewing road representation learning as a mere feature engineering task, we formulate it as a *spectral disentanglement problem*

on graph signals. Guided by the core philosophy of "seeking commonality while preserving specificity," our goal is to learn a mapping $f_\theta$ that projects disparate road networks into a unified latent manifold. Formally, we decompose the segment signal $\mathbf{X}_S$ into two orthogonal subspaces: a *low-frequency subspace* capturing city-invariant functional patterns (Commonality), and a *high-frequency subspace* absorbing city-specific geometric variations (Specificity).

As shown in Figure 2, CoSpec consists of two theoretically grounded modules: *Manifold Alignment via Adaptive Structure Learning* (Sec. 4.1), which establishes a region graph to bound the Wasserstein distance between cities; and *Spectral-Aware Dual-Path Reconstruction* (Sec. 4.2), which separates functional flow from geometric noise; *Regularized Optimization* (Sec. 4.3) introduces the training objective, which integrates EM-based prototype learning and regularization to improve representation quality.

### 4.1. Manifold Alignment via Adaptive Structure Learning

**Why Region Abstraction?** A major obstacle in cross-city transfer is the *geometric distribution shift* at the segment level (e.g., varying road densities). Aligning segments directly is ill-posed. However, we argue that *functional semantics are locally smooth and thus stable under spatial aggregation*, which motivates learning a coarse-grained region manifold as the anchor for cross-city alignment. We rigorously justify constructing a coarse-grained region graph $\mathcal{G}_R$ via the following contraction theorem:

**Theorem 4.1** (Wasserstein Contraction). *Let $\phi : \mathcal{G}_S \to \mathcal{G}_R$ be the aggregation mapping defined by the assignment matrix $\mathbf{A}_{SR}$. If $\phi$ is Lipschitz continuous (satisfied by our bounded soft assignment), then the mapping is a contraction on the probability space. Specifically, for any two cities $A$ and $B$, the 2-Wasserstein distance between their distributions at the region level is upper-bounded by that at the segment level:*

$$W_2(\mu_{region}^{(A)}, \mu_{region}^{(B)}) \leq W_2(\mu_{segment}^{(A)}, \mu_{segment}^{(B)}). \quad (1)$$

*Proof Sketch.* The aggregation operator acts as a 1-Lipschitz map. By the dual formulation of optimal transport, applying a Lipschitz map to distributions contracts the Wasserstein distance. See **Appendix A** for details.

Guided by Theorem 4.1, our first module aims to learn this optimal mapping $\mathbf{A}_{SR}$ to minimize domain discrepancy.

#### 4.1.1. SEGMENT CONTEXTUAL ENCODING

Road attributes, such as road type (RT), lane number (LN), and discretized length (DL), are discrete and heterogeneous. To transform them into a continuous, homogeneous graph signal suitable for spectral processing, we employ a multi-

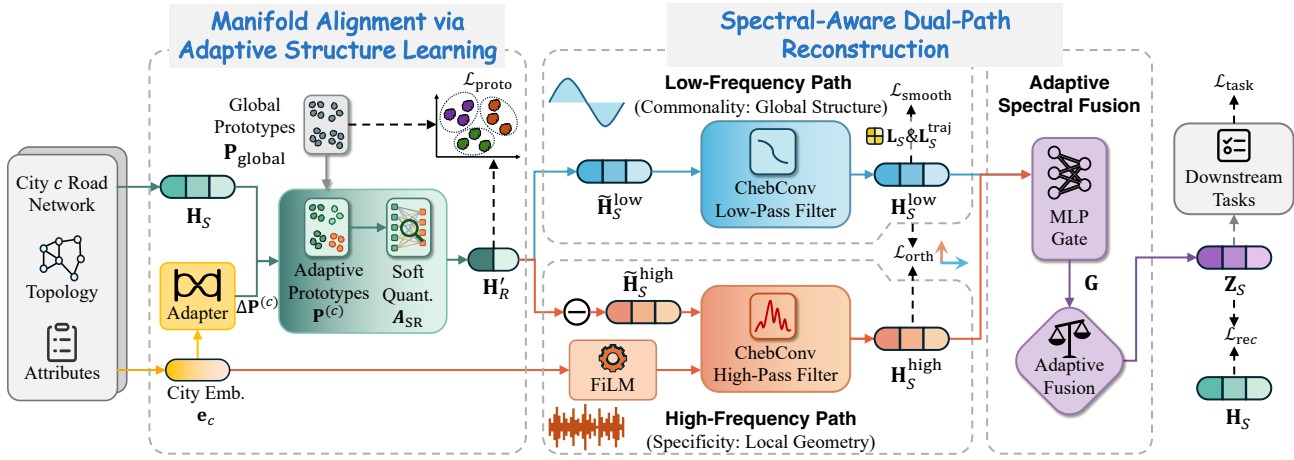

*Figure 2.* The overall framework of CoSpec.

modal embedding layer.

Let $\mathbf{x}_i$ be the attribute tuple of segment $s_i$. We obtain the initial signal $\mathbf{h}_i^{(0)}$ through

$$\mathbf{h}_i^{(0)} = \mathbf{W}_{\text{proj}}(\mathbf{E}_{\text{RT}}[x_i^{\text{RT}}] \oplus \mathbf{E}_{\text{LN}}[x_i^{\text{LN}}] \oplus \mathbf{E}_{\text{DL}}[x_i^{\text{DL}}]) + \mathbf{b}_{\text{proj}}, \quad (2)$$

where $\mathbf{E}_*[\cdot]$ are learnable lookup tables for each attribute type, $\oplus$ denotes concatenation, and $\mathbf{W}_{\text{proj}} \in \mathbb{R}^{d \times d_{\text{in}}}, \mathbf{b}_{\text{proj}} \in \mathbb{R}^d$ are projection parameters.

Since road segments exhibit strong spatial autocorrelation, we employ a GCN (Kipf & Welling, 2017) as a local contextual encoder. To incorporate structural bias while preventing the loss of high-frequency details (over-smoothing), we utilize a shallow aggregation with residual connections:

$$\mathbf{H}_S = \text{LayerNorm}\left(\text{GCN}(\mathbf{H}^{(0)}, \mathbf{A}_S) + \mathbf{H}^{(0)}\right), \quad (3)$$

where $\mathbf{H}^{(0)} \in \mathbb{R}^{N_S \times d}$ denotes the initial feature matrix derived from attribute embeddings.

### 4.1.2. LATENT QUANTIZATION VIA PROTOTYPE LEARNING

To align road networks from disparate distributions, we require a shared semantic space. We formulate this as a *Dictionary Learning* problem, aiming to discover a set of canonical functional bases (*prototypes*) that constitute a universal vocabulary for urban semantics. Intuitively, these prototypes represent distinct functional categories shared across cities, such as "Commercial Hubs", "Residential Zones", or "Industrial Districts".

**City-Adaptive Basis Generation.** However, a rigid global dictionary is insufficient due to cross-city domain shifts. For instance, the geometric layout of a downtown area in a dense city differs significantly from one in a sprawling city. To reconcile this functional universality with local geometric specificity, we propose a *Residual Adaptation Mechanism*.

Let $\mathbf{P}_{\text{global}} \in \mathbb{R}^{N_R \times d}$ be the learnable global dictionary. We dynamically deform these bases to fit the local manifold of city $c$ via a low-rank residual adapter:

$$\mathbf{P}^{(c)} = \mathbf{P}_{\text{global}} + \underbrace{\mathbf{W}_{\text{up}}(\sigma(\mathbf{W}_{\text{down}}\mathbf{e}_c))}_{\Delta \mathbf{P}^{(c)}}, \quad (4)$$

where $\mathbf{e}_c$ is the learnable city embedding. The bottleneck structure ($\mathbf{W}_{\text{down}} \in \mathbb{R}^{r \times d}, \mathbf{W}_{\text{up}} \in \mathbb{R}^{d \times r}, r \ll d$) ensures that the adaptation $\Delta \mathbf{P}^{(c)}$ captures only the necessary shifts without corrupting the global semantic structure. This design prevents negative transfer caused by enforcing rigid prototype alignment across cities with incompatible geometries, while still preserving a shared semantic anchor space.

**Soft Quantization.** With the adapted dictionary $\mathbf{P}^{(c)}$, we project the segment features $\mathbf{H}_S$ onto this semantic coordinate system. We compute the soft assignment $\mathbf{A}_{SR} \in \mathbb{R}^{N_S \times N_R}$ via temperature-scaled attention:

$$\mathbf{A}_{SR}[i,j] = \frac{\exp(\langle \mathbf{h}_{S,i}\mathbf{W}_Q^{\text{pool}}, \mathbf{p}_j^{(c)}\mathbf{W}_K^{\text{pool}}\rangle/\tau)}{\sum_{k=1}^{N_R} \exp(\langle \mathbf{h}_{S,i}\mathbf{W}_Q^{\text{pool}}, \mathbf{p}_k^{(c)}\mathbf{W}_K^{\text{pool}}\rangle/\tau)}, \quad (5)$$

where $\mathbf{W}_Q^{\text{pool}}, \mathbf{W}_K^{\text{pool}}$ are learnable projections, and $\tau$ denotes the temperature. This process effectively performs a *soft vector quantization*, clustering geometrically diverse segments into semantically aligned functional regions.

### 4.1.3. REGION GRAPH INDUCTION

Based on the assignment $\mathbf{A}_{SR}$, we lift the microscopic segment signals to the macroscopic region manifold. The region features $\mathbf{H}_R$ and topology $\mathbf{A}_R$ are constructed as

$$\mathbf{H}_R = \mathbf{A}_{SR}^\top \mathbf{H}_S + \mathbf{P}^{(c)}, \quad \mathbf{A}_R = \mathbf{A}_{SR}^\top \mathbf{A}_S \mathbf{A}_{SR}. \quad (6)$$

Here, adding the prototypes $\mathbf{P}^{(c)}$ serves as a *semantic residual connection*, ensuring that region representations remain anchored to the canonical functional bases even if the aggregated segment features are noisy.

To model long-range interactions, we apply a Graph Transformer on $\mathcal{G}_R$. The region representations are updated via multi-head self-attention with a structural bias:

$$\mathbf{H}'_R = \text{softmax}\left(\frac{(\mathbf{H}_R\mathbf{W}_Q)(\mathbf{H}_R\mathbf{W}_K)^\top}{\sqrt{d}} + \lambda\mathbf{A}_R\right)(\mathbf{H}_R\mathbf{W}_V). \tag{7}$$

Here, the adjacency matrix $\mathbf{A}_R$ is added directly to the attention logits, serving as a hard structural bias to prioritize physically connected regions.

### 4.2. Spectral-Aware Dual-Path Reconstruction

To explicitly disentangle commonality from specificity, we adopt the perspective of *Graph Signal Processing* (Shuman et al., 2013; Ortega et al., 2018): spatially smooth functional patterns correspond to the *low-frequency spectrum*, while local geometric variations dominate the *high-frequency spectrum*. While the region abstraction in Sec. 4.1 effectively aligns cross-city distributions (Theorem 4.1), it introduces a side effect: the loss of local details. Inspired by (Ma et al., 2026), we formally quantify this information loss via the following theorem:

**Theorem 4.2** (Hierarchical Abstraction as Low-Pass Filtering). *Let $\mathbf{L}_S$ be the Laplacian of the segment graph. The hierarchical loop operator $\mathbf{P} = \mathbf{A}_{SR}\mathbf{A}_{SR}^\top$ acts as a spectral contraction mapping. For any graph signal $\mathbf{z}$, its Dirichlet energy satisfies*

$$(\mathbf{Pz})^\top\mathbf{L}_S(\mathbf{Pz}) \leq \mathbf{z}^\top\mathbf{L}_S\mathbf{z}. \tag{8}$$

*This inequality implies that projecting segments to regions inherently acts as a **spectral low-pass filter**, attenuating high-frequency components (specificity) while preserving low-frequency smoothness (commonality).*

*Proof Sketch.* The operator $\mathbf{P}$ corresponds to local averaging within soft clusters. In the spectral domain, averaging suppresses eigenvectors associated with large eigenvalues (high frequencies). See **Appendix A** for details.

Theorem 4.2 reveals that relying solely on region-level features would result in over-smoothed representations lacking local specificity. To resolve this, we propose a Dual-Path Reconstruction Architecture: (1) A *Low-Frequency Path* to reconstruct the global skeleton from regions; and (2) A dedicated *High-Frequency Path* to explicitly recover and adapt the lost geometric residuals.

#### 4.2.1. Low-Frequency Path: Global Consistency Reconstruction

This path aims to reconstruct the city-invariant functional backbone by smoothing the coarse-grained region signals. We first map the learned region representations back to the segment domain:

$$\tilde{\mathbf{H}}_S^{\text{low}} = \mathbf{A}_{SR}\mathbf{H}'_R. \tag{9}$$

Mathematically, this up-projection functions as a *Zero-order Hold interpolation* on the graph. While efficient, this operation introduces sharp discontinuities at region boundaries, generating high-frequency harmonics known as *spectral imaging artifacts* (Shuman et al., 2013; Ortega et al., 2018).

To eliminate these artifacts and recover the underlying smooth manifold, we require a filter with a large receptive field that strictly limits high-frequency oscillations. We employ Chebyshev Graph Convolution (ChebConv) (Defferrard et al., 2016) as a learnable spectral low-pass filter:

$$\mathbf{H}_S^{\text{low}} = \sum_{k=0}^{K_L}\mathbf{T}_k(\tilde{\mathbf{L}}_S)\tilde{\mathbf{H}}_S^{\text{low}}\mathbf{\Theta}_k^{\text{low}}. \tag{10}$$

Here, $\tilde{\mathbf{L}}_S = 2\mathbf{L}_S/\lambda_{\max} - \mathbf{I}$ is the scaled Laplacian ($\lambda_{\max}$ is the largest eigenvalue), and $\mathbf{T}_k(x) = 2x\mathbf{T}_{k-1}(x) - \mathbf{T}_{k-2}(x)$ denotes the Chebyshev polynomials of order $k$.

*Why ChebConv?* Unlike standard GCNs that are limited to 1-hop aggregation, a polynomial filter of order $K_L$ approximates the heat diffusion kernel over a $K_L$-hop radius. This allows us to explicitly control the smoothing scale, effectively "melting" the block boundaries while preserving the global functional layout.

#### 4.2.2. High-Frequency Path: Adaptive Residual Modulation

While the low-frequency path recovers the global skeleton, it inherently smooths out local geometric details (e.g., specific road curvature or intersection complexity). This path targets the *Spectral Complement*—the high-frequency specificity.

**Residual Extraction via Spectral Subtraction.** We isolate the raw high-frequency signal $\tilde{\mathbf{H}}_S^{\text{high}}$ by subtracting the reconstructed low-frequency backbone from the original signal, analogous to a Laplacian Pyramid decomposition (Burt & Adelson, 1987):

$$\tilde{\mathbf{H}}_S^{\text{high}} = \mathbf{H}_S - \tilde{\mathbf{H}}_S^{\text{low}}. \tag{11}$$

This residual term captures the *deviation* of each segment from its regional mean. Mathematically, it absorbs the high-frequency energy that was attenuated by the Zero-order Hold and ChebConv operations in the parallel path.

**Geometric Style Adaptation.** Crucially, these geometric residuals exhibit the most severe domain shifts (*e.g.,* the "texture" of different road networks). To align these *geometric styles*, we employ Feature-wise Linear Modulation (FiLM) (Perez et al., 2018). This acts as a conditional normalization that recalibrates the statistics of the residuals based on the city embedding $\mathbf{e}_c$ (see Eq. (4)):

$$\hat{\mathbf{H}}_S^{\text{high}} = (\mathbf{1} + \boldsymbol{\gamma}_c(\mathbf{e}_c)) \odot \tilde{\mathbf{H}}_S^{\text{high}} + \boldsymbol{\beta}_c(\mathbf{e}_c). \tag{12}$$

Here, the coefficients $\boldsymbol{\gamma}_c, \boldsymbol{\beta}_c \in \mathbb{R}^d$ perform an affine transformation to map the city-specific residual distribution to a canonical latent space, enabling positive transfer.

**Topology-Constrained High-Pass Filtering.** Since FiLM is a node-wise operation, it ignores local connectivity. To ensure the adapted residuals respect the graph topology, we apply a spectral high-pass filter using ChebConv with a small receptive field ($K_H$):

$$\mathbf{H}_S^{\text{high}} = \sum_{k=0}^{K_H} \mathbf{T}_k(\tilde{\mathbf{L}}_S) \hat{\mathbf{H}}_S^{\text{high}} \mathbf{\Theta}_k^{\text{high}}. \qquad (13)$$

Unlike the low-frequency path, here we use a small $K_H$ (e.g., $K_H = 1$ or $2$). This strictly limits the diffusion range, preserving the *locality* of the geometric features and preventing them from over-smoothing into global patterns.

### 4.2.3. ADAPTIVE SPECTRAL FUSION

Road segments exhibit distinct spectral signatures depending on their functional roles. To capture this topological diversity, we employ a spatially adaptive gate $\mathbf{G} \in \mathbb{R}^{N_S \times d}$ to dynamically balance the spectral components:

$$\mathbf{G} = \sigma \left( \mathbf{W}_{\text{gate}} [\mathbf{H}_S^{\text{low}} \oplus \mathbf{H}_S^{\text{high}}] + \mathbf{b}_{\text{gate}} \right), \qquad (14)$$

$$\mathbf{Z}_S = \mathbf{G} \odot \mathbf{H}_S^{\text{low}} + (\mathbf{1} - \mathbf{G}) \odot \mathbf{H}_S^{\text{high}}, \qquad (15)$$

where $\mathbf{W}_{\text{gate}}, \mathbf{b}_{\text{gate}}$ denote learnable weights and $\sigma$ is the sigmoid function. $\mathbf{G}$ acts as a learnable spectral filter that operates element-wise. It allows the model to automatically suppress noise in the high-frequency band for smooth arterial roads, while preserving sharp geometric details for complex local neighborhoods.

### 4.3. Regularized Optimization

We train CoSpec end-to-end using a composite objective function that enforces the structural and spectral properties postulated above.

**Signal Reconstruction Loss ($\mathcal{L}_{\text{rec}}$).** To ensure the decomposition is lossless and the representation preserves input information, we minimize the reconstruction error:

$$\mathcal{L}_{\text{rec}} = \frac{1}{N_S} \|\mathbf{H}_S - \mathbf{Z}_S\|_F^2. \qquad (16)$$

**Spectral Orthogonality Loss ($\mathcal{L}_{\text{orth}}$).** A critical requirement for disentanglement is that the two subspaces must be independent. We enforce strict orthogonality using column-wise cosine similarity:

$$\mathcal{L}_{\text{orth}} = \left\| \bar{\mathbf{H}}_S^{\text{low}\top} \bar{\mathbf{H}}_S^{\text{high}} \right\|_F^2, \quad \text{where } \bar{\mathbf{H}} = \frac{\mathbf{H}}{\|\mathbf{H}\|_{\text{col}}}. \qquad (17)$$

Minimizing this ensures that the high-frequency path does not redundantly learn global patterns (Information Leakage).

**Graph Smoothness Loss ($\mathcal{L}_{\text{smooth}}$).** We enforce the smoothness prior on the low-frequency component by minimizing

its Dirichlet energy on both the static topology graph $\mathbf{L}_S$ and the dynamic trajectory graph $\mathbf{L}_S^{\text{traj}}$:

$$\mathcal{L}_{\text{smooth}} = \text{Tr} \left( \mathbf{H}_S^{\text{low}\top} \mathbf{L}_S \mathbf{H}_S^{\text{low}} \right) + \alpha \cdot \text{Tr} \left( \mathbf{H}_S^{\text{low}\top} \mathbf{L}_S^{\text{traj}} \mathbf{H}_S^{\text{low}} \right). \qquad (18)$$

This explicitly penalizes high-frequency oscillations in the commonality branch, ensuring spatial contiguity.

**Prototype Alignment Loss ($\mathcal{L}_{\text{proto}}$).** To align the learned regions with the canonical functional bases, we adopt an iterative *Pseudo-Labeling* strategy inspired by Expectation-Maximization (EM).

*E-step (Assignment):* We assign each region $r_i$ to its nearest adapted prototype to generate a pseudo-label $k^*$:

$$k^*(r_i) = \arg \min_{k \in \{1, \dots, K_P\}} \|\mathbf{h}_{R,i} - \mathbf{p}_k^{(c)}\|_2. \qquad (19)$$

*M-step (Optimization):* We optimize the encoder parameters by maximizing the mutual information via InfoNCE loss:

$$\mathcal{L}_{\text{proto}} = -\frac{1}{N_R} \sum_{i=1}^{N_R} \log \frac{\exp(\mathbf{h}_{R,i} \cdot \mathbf{p}_{k^*}^{(c)}/\tau)}{\sum_{j=1}^{K_P} \exp(\mathbf{h}_{R,i} \cdot \mathbf{p}_j^{(c)}/\tau)}. \qquad (20)$$

This effectively pulls regions towards their functional centers while pushing away from mismatched prototypes.

**Total Objective.** The final loss is a weighted sum:

$$\mathcal{L}_{\text{total}} = \lambda_1 \mathcal{L}_{\text{rec}} + \lambda_2 \mathcal{L}_{\text{orth}} + \lambda_3 \mathcal{L}_{\text{smooth}} + \lambda_4 \mathcal{L}_{\text{proto}}, \qquad (21)$$

where $\lambda_*$ are hyperparameters balancing the trade-off.

## 5. Experiments

In this section, we conduct experiments to demonstrate the effectiveness of our proposed model.

### 5.1. Experimental Setup

#### 5.1.1. CONSTRUCTION OF THE DATASETS

To evaluate the cross-city generalization ability of our model, we utilize large-scale datasets from three metropolises: Beijing (*BJ*), Chengdu (*CD*), and Xi'an (*XA*). These cities exhibit distinct road network topologies: *BJ* features a classic grid-like layout, while *CD* is characterized by a concentric ring structure, and *XA* presents a mixed grid-irregular pattern. Road networks are extracted from *OpenStreetMap* [1]. We standardize the preprocessing pipeline for trajectory data following the protocol in HRNR (Wu et al., 2020). Detailed descriptions of the datasets are summarized in Appendix B.

#### 5.1.2. BASELINES

In our experiments, we consider three categories of baselines for a comprehensive comparison:

---

[1] https://www.openstreetmap.org/

*Table 1.* Overall performance comparison across four tasks. **Bold** denotes the best result, and underline denotes the runner-up.

| Task | Dataset | Metric | DW | IRN2Vec | Toast | GeomGCN | HRNR | GT | NFormer | GCN | FAGCN | ChebNetII | HiFiNet | CoSpec |
|---|---|---|---|---|---|---|---|---|---|---|---|---|---|---|
| Next Location Prediction | BJ | ACC@1↑ | 0.383 | 0.371 | 0.391 | 0.391 | 0.412 | 0.362 | 0.371 | 0.387 | 0.395 | 0.407 | 0.426 | **0.438** |
| | | ACC@5↑ | 0.527 | 0.498 | 0.542 | 0.526 | 0.556 | 0.483 | 0.502 | 0.517 | 0.554 | 0.573 | 0.587 | **0.615** |
| | CD | ACC@1↑ | 0.403 | 0.324 | 0.369 | 0.398 | 0.420 | 0.379 | 0.381 | 0.388 | 0.397 | 0.382 | 0.442 | **0.456** |
| | | ACC@5↑ | 0.556 | 0.454 | 0.542 | 0.546 | 0.571 | 0.506 | 0.515 | 0.552 | 0.591 | 0.580 | 0.665 | **0.677** |
| | XA | ACC@1↑ | 0.346 | 0.324 | 0.335 | 0.346 | 0.376 | 0.316 | 0.318 | 0.333 | 0.393 | 0.386 | 0.399 | **0.410** |
| | | ACC@5↑ | 0.461 | 0.457 | 0.460 | 0.476 | 0.500 | 0.456 | 0.475 | 0.455 | 0.519 | 0.523 | 0.546 | **0.587** |
| Label Classification | BJ | F1↑ | 0.676 | 0.733 | 0.679 | 0.773 | 0.819 | 0.821 | 0.823 | 0.790 | 0.832 | 0.815 | 0.838 | **0.851** |
| | | AUC↑ | 0.825 | 0.836 | 0.825 | 0.846 | 0.885 | 0.883 | 0.887 | 0.849 | 0.901 | 0.891 | 0.906 | **0.922** |
| | CD | F1↑ | 0.702 | 0.686 | 0.645 | 0.719 | 0.716 | 0.763 | 0.772 | 0.716 | 0.798 | 0.800 | 0.796 | **0.809** |
| | | AUC↑ | 0.721 | 0.706 | 0.712 | 0.735 | 0.782 | 0.805 | 0.835 | 0.734 | 0.841 | 0.861 | 0.869 | **0.879** |
| | XA | F1↑ | 0.626 | 0.627 | 0.640 | 0.651 | 0.694 | 0.705 | 0.701 | 0.646 | 0.714 | 0.689 | 0.720 | **0.756** |
| | | AUC↑ | 0.639 | 0.629 | 0.653 | 0.665 | 0.716 | 0.736 | 0.733 | 0.661 | 0.803 | 0.758 | 0.811 | **0.836** |
| Destination Prediction | BJ | ACC@1↑ | 0.229 | 0.215 | 0.271 | 0.242 | 0.277 | 0.273 | 0.275 | 0.232 | 0.281 | 0.276 | 0.297 | **0.309** |
| | | ACC@5↑ | 0.321 | 0.313 | 0.396 | 0.357 | 0.401 | 0.396 | 0.399 | 0.352 | 0.425 | 0.406 | 0.428 | **0.453** |
| | CD | ACC@1↑ | 0.187 | 0.235 | 0.239 | 0.267 | 0.281 | 0.282 | 0.284 | 0.251 | 0.266 | 0.253 | 0.295 | **0.312** |
| | | ACC@5↑ | 0.362 | 0.346 | 0.342 | 0.394 | 0.407 | 0.405 | 0.409 | 0.377 | 0.401 | 0.409 | 0.426 | **0.438** |
| | XA | ACC@1↑ | 0.167 | 0.210 | 0.198 | 0.226 | 0.256 | 0.254 | 0.260 | 0.217 | 0.302 | 0.283 | 0.291 | **0.310** |
| | | ACC@5↑ | 0.289 | 0.305 | 0.306 | 0.352 | 0.375 | 0.364 | 0.379 | 0.334 | 0.436 | 0.418 | 0.430 | **0.447** |

**Random Walk-based Models**: These methods learn node embeddings by generating stochastic walk sequences and applying shallow embedding techniques. Representative baselines include *DW* (Perozzi et al., 2014), *IRN2Vec* (Wang et al., 2019b), and *Toast* (Chen et al., 2021).

**Spatial-based GNNs**: These baselines aggregate information directly in the spatial domain, leveraging geometric structures or attention mechanisms for feature learning. We consider *GeomGCN* (Pei et al., 2020), *HRNR* (Wu et al., 2020), and transformer architectures *GT* (Dwivedi & Bresson, 2021) and *NFormer* (Wu et al., 2022).

**Spectral-based GNNs**: These methods operate in the graph frequency domain to design learnable spectral filters. We include the foundational *GCN* (Kipf & Welling, 2017), advanced *FAGCN* (Bo et al., 2021) and *ChebNetII* (He et al., 2022), and the recent SOTA *HiFiNet* (Ma et al., 2026).

Detailed adaptation strategies are discussed in Appendix C.

### 5.1.3. EVALUATION TASKS AND METRICS

We evaluate the learned representations on three downstream tasks: (1) *Next Location Prediction*, assessing local transition patterns; (2) *Destination Prediction*, testing long-range intent inference; and (3) *Label Classification*, evaluating the preservation of static semantic attributes (*e.g.,* bridge, tunnel). For the two prediction tasks, we employ *ACC@1* and *ACC@5* to measure hit rates. For label classification, we report *F1-score* and *AUC* to assess performance. Detailed setups are provided in Appendix D.

### 5.2. Results and Analysis

Table 1 reports the performance of CoSpec and baseline methods across three datasets. CoSpec consistently achieves the best results on all downstream tasks, demonstrating the

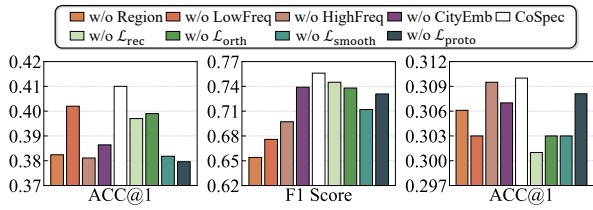

(a) Next Location Prediction (b) Label Classification (c) Destination Prediction

*Figure 3.* Ablation study results on the XA dataset.

effectiveness of our framework. We further analyze the results by grouping baselines into three categories.

Random walk–based methods generally yield the weakest performance. Although *Toast* benefits from incorporating traffic context, these approaches rely on shallow embedding schemes derived from stochastic local sequences, which limits their ability to capture long-range dependencies and hierarchical structures in large-scale road networks.

Spatial-based GNNs show competitive performance on tasks dominated by local structural patterns. For example, *HRNR* performs well on Next Location Prediction due to its hierarchical pooling design. However, these methods, including advanced Graph Transformers such as *NodeFormer*, tend to underperform on global tasks like Destination Prediction. This limitation likely stems from their reliance on localized aggregation, which can lead to over-smoothing and reduced sensitivity to global topology.

Spectral-based GNNs form a strong comparison group. Models such as *FAGCN* and *ChebNetII* outperform *GCN* by enabling adaptive or higher-order filtering, while *HiFiNet* further improves performance through hierarchical frequency decomposition. Nevertheless, CoSpec consistently surpasses all spectral baselines. Unlike standard spectral methods that learn filters tightly coupled to the Laplacian

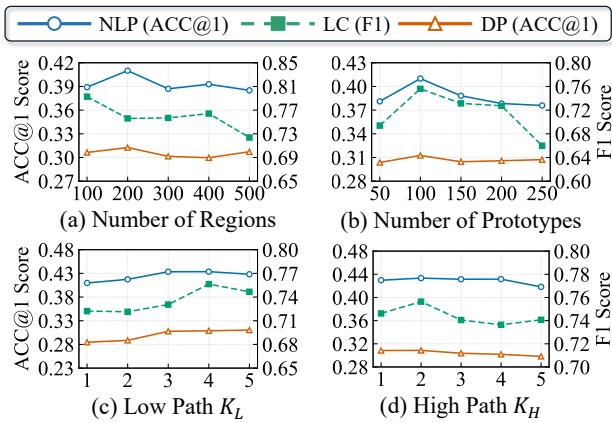

Figure 4. Parameter sensitivity analysis on the XA dataset.

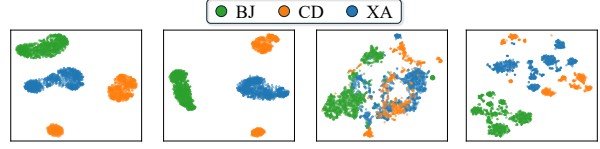

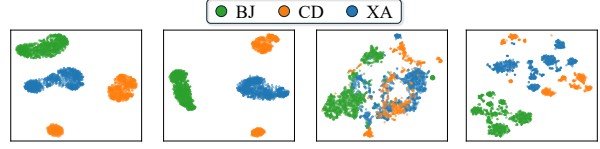
(a) HiFiNet: L-Freq (b) HiFiNet: H-Freq (c) CoSpec: L-Freq (d) CoSpec: H-Freq

*Figure 5.* t-SNE visualization of disentangled representations.

spectrum of a specific city, CoSpec explicitly separates shared spectral patterns from city-specific variations, leading to more transferable and robust representations.

### 5.3. Ablation Study

To assess the contribution of each component in CoSpec, we conduct an ablation study by selectively removing individual modules. Results on the XA dataset are reported in Figure 3, with consistent trends observed across other datasets. The full model consistently achieves the best performance.

**Structural components.** Removing the Region Embedding (*w/o Region*) leads to substantial performance degradation across all tasks. Without regions, the model fails to separate global common patterns from local specific variations. Additionally, removing the City Embedding (*w/o CityEmb*) particularly degrades Next Location Prediction, highlighting the necessity of modeling city-specific semantics under cross-city distribution shifts.

**Spectral components.** Different tasks exhibit distinct dependencies on frequency. Excluding low-frequency components (*w/o LowFreq*) causes the largest drops in Label Classification and Destination Prediction, confirming that low-frequency signals encode global functional semantics. In contrast, removing high-frequency components (*w/o HighFreq*) severely impacts Next Location Prediction, which relies on fine-grained local structural details.

**Auxiliary objectives.** Removing either the orthogonality loss (*w/o $\mathcal{L}_{orth}$*) or the reconstruction loss (*w/o $\mathcal{L}_{rec}$*) consistently degrades performance, demonstrating the necessity of explicit disentanglement and information preservation. Similarly, performance drops under *w/o $\mathcal{L}_{smooth}$* and *w/o $\mathcal{L}_{proto}$* validate the importance of enforcing graph smoothness and prototype alignment for structural consistency and discriminative learning.

### 5.4. Parameter Sensitivity

We investigate the sensitivity of CoSpec to key hyperparameters on the Xi'an dataset, as illustrated in Figure 4.

First, regarding the structural parameters, both the number of regions ($N_R$) and prototypes ($N_P$) exhibit a distinct bell-shaped trend. Specifically, performance generally peaks at $N_R = 200$, balancing the trade-off between coarse global semantics and fine-grained local resolution. Similarly, the number of prototypes reaches an optimal point at $N_P = 100$; fewer prototypes fail to cover diverse functional semantics, while excessive ones lead to redundancy and overfitting. Second, regarding the spectral components, we observe distinct sensitivities to frequency bandwidths. For the low-frequency path ($K_L$), performance improves with more components and saturates around 3 or 4, confirming that sufficient low-frequency bases are necessary to capture global commonalities. In contrast, the high-frequency path ($K_H$) favors a smaller number of components (peaking at 2), indicating that while leading high-frequency signals encode valuable local specificities, including too many introduces noise that degrades representation quality.

### 5.5. Qualitative Analysis

To validate the disentanglement effectiveness, we visualize the learned road representations of Beijing, Chengdu, and Xi'an using t-SNE (Maaten & Hinton, 2008). Figure 5 reveals that the baseline HiFiNet struggles to bridge significant inter-city discrepancies, as its embeddings form disjoint clusters strongly correlated with city identities. This clear separation suggests that the learned features overfit to local spatial layouts, evidenced by the distinct ring-like shape retained in the Chengdu cluster. In sharp contrast, CoSpec achieves a high degree of alignment where the low-frequency distributions are deeply interwoven into a shared semantic manifold, confirming its ability to filter out topological noise. Furthermore, CoSpec displays more granular clustering in the high-frequency space compared to HiFiNet, indicating superior capability in capturing fine-grained local details while successfully isolating common knowledge.

## 6. Conclusion

In this paper, we tackled the challenge of *Unified Cross-City Road Representation Learning*, identifying spectral misalignment as the core obstacle preventing GNNs from generalizing across heterogeneous urban topologies. We proposed CoSpec, a novel framework that fundamentally disentangles road networks into universally shareable functional commonalities and city-unique geometric specificities. By implementing a spectral-aware dual-path architecture, CoSpec aligns macro-level semantics through region-based

prototypes while adaptively modulating high-frequency local residuals. Theoretical analysis confirms that this hierarchical abstraction effectively bounds the distribution shift, and extensive experiments on real-world datasets demonstrate CoSpec's superiority over state-of-the-art baselines. Future work will extend this framework to incorporate dynamic temporal signals and explore zero-shot transfer capabilities for unseen cities.

## Acknowledgements

Prof. Wang's work was partially supported by the Science and Technology Development Fund Macau SAR (0052/2023/RIA1), National Natural Science Foundation of China (No. 72242101, 72222022, 72171013), and the Special Fund for Health Development Research of Beijing (2024-2G-30121). Prof. U's work was partially supported by the Science and Technology Development Fund Macau SAR (0003/2023/RIC, 0052/2023/RIA1, 0011/2025/RIC, 001/2024/SKL for SKL-IOTSC).

## Impact Statement

This work aims to advance machine learning for unified cross-city road representation learning. More broadly, it studies how knowledge can be transferred across heterogeneous urban structures through a shared prototype space while preserving city-specific variations. By disentangling city-invariant functional patterns from city-specific geometric structures, the proposed framework may improve the scalability and transferability of urban intelligence systems, especially in data-scarce cities where training city-specific models is costly.

The potential applicability of this idea beyond road networks should be considered with clear boundary conditions. Its effectiveness depends on whether the target urban domain contains sufficiently strong cross-city commonalities that can be represented by a compact prototype space. Moreover, road segments have relatively consistent attributes across cities, whereas other urban intelligence tasks may involve more heterogeneous entities and modalities, requiring additional alignment mechanisms. Our method relies on historical road network and trajectory data and may inherit biases from such data, but it does not introduce new data sources or surveillance mechanisms and operates on anonymized, aggregated information commonly used in prior research. With appropriate data governance and human oversight, we believe its benefits in improving scalable cross-city learning outweigh the foreseeable risks.

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

# A. Proofs of Theoretical Results

This appendix provides detailed proofs for Theorem 4.1 (Wasserstein contraction) and Theorem 4.2 (hierarchical abstraction as low-pass filtering). Throughout, we focus on the segment-to-region assignment matrix $\mathbf{A}_{SR} \in \mathbb{R}^{N_S \times N_R}$ produced by Eq. (5) in the main paper.

## A.1. Notation and Basic Assumptions

Let $\mathbf{H}_S \in \mathbb{R}^{N_S \times d}$ denote segment embeddings (or any graph signal), and define region embeddings by the linear aggregation

$$\mathbf{H}_R = \mathbf{A}_{SR}^{\top} \mathbf{H}_S. \tag{22}$$

We assume $\mathbf{A}_{SR}$ is *row-stochastic* (soft assignment):

$$\mathbf{A}_{SR}[i, j] \geq 0, \qquad \sum_{j=1}^{N_R} \mathbf{A}_{SR}[i, j] = 1, \ \ \forall i. \tag{23}$$

This is satisfied by the softmax in Eq. (5).

We will use the normalized Frobenius metric on node-sets:

$$\|\mathbf{H}\|_{F,\text{avg}} = \left( \frac{1}{N} \|\mathbf{H}\|_F^2 \right)^{1/2}. \tag{24}$$

## A.2. Wasserstein Contraction via 1-Lipschitz Aggregation

We first prove that the aggregation mapping in Eq. (22) is non-expansive.

**Lemma A.1** (Aggregation is 1-Lipschitz). *Assume* $\mathbf{A}_{SR}$ *satisfies Eq.* (23). *Then the linear map* $\Phi(\mathbf{H}_S) = \mathbf{A}_{SR}^{\top} \mathbf{H}_S$ *is* 1-*Lipschitz under* $\| \cdot \|_{F,\text{avg}}$*, i.e.,*

$$\|\mathbf{A}_{SR}^{\top} \mathbf{H} - \mathbf{A}_{SR}^{\top} \mathbf{H}'\|_{F,\text{avg}} \leq \|\mathbf{H} - \mathbf{H}'\|_{F,\text{avg}}.$$

*Proof.* Let $\Delta = \mathbf{H} - \mathbf{H}'$ and consider one feature dimension (column) $\delta \in \mathbb{R}^{N_S}$. For each region $r$, $(\mathbf{A}_{SR}^{\top} \delta)_r = \sum_i \mathbf{A}_{SR}[i, r] \delta_i$ is a convex combination of $\{\delta_i\}$. By Jensen's inequality,

$$(\mathbf{A}_{SR}^{\top} \delta)_r^2 = \left( \sum_i \mathbf{A}_{SR}[i, r] \delta_i \right)^2 \leq \sum_i \mathbf{A}_{SR}[i, r] \delta_i^2.$$

Summing over $r$ and swapping sums,

$$\|\mathbf{A}_{SR}^{\top} \delta\|_2^2 \leq \sum_r \sum_i \mathbf{A}_{SR}[i, r] \delta_i^2 = \sum_i \left( \sum_r \mathbf{A}_{SR}[i, r] \right) \delta_i^2 = \|\delta\|_2^2,$$

where we used row-stochasticity. Applying the same argument to all $d$ feature dimensions yields $\|\mathbf{A}_{SR}^{\top} \Delta\|_F \leq \|\Delta\|_F$. After normalization by $N_S$ and $N_R$, the inequality remains non-expansive under $\| \cdot \|_{F,\text{avg}}$. $\square$

**Lemma A.2** (Wasserstein contraction under Lipschitz maps). *Let* $f : \mathcal{X} \to \mathcal{Y}$ *be an L-Lipschitz map between Euclidean spaces, and let* $\mu, \nu$ *be probability measures on* $\mathcal{X}$ *with finite second moments. Then*

$$W_2(f_\# \mu, f_\# \nu) \leq L \, W_2(\mu, \nu).$$

*Proof.* Recall that $\Pi(\mu, \nu)$ denotes the set of couplings of $\mu$ and $\nu$, i.e., probability measures $\gamma$ on $\mathcal{X} \times \mathcal{X}$ whose marginals are $\mu$ and $\nu$. The 2-Wasserstein distance is

$$W_2^2(\mu, \nu) = \inf_{\gamma \in \Pi(\mu, \nu)} \int_{\mathcal{X} \times \mathcal{X}} \|x - y\|^2 \, d\gamma(x, y).$$

Take any $\gamma \in \Pi(\mu, \nu)$. Define a measure $\tilde{\gamma}$ on $\mathcal{Y} \times \mathcal{Y}$ as the pushforward of $\gamma$ by the map $(f, f)$:

$$\tilde{\gamma} := (f, f)_\# \gamma, \qquad \text{i.e.,} \quad \tilde{\gamma}(B) = \gamma\big((f, f)^{-1}(B)\big) \text{ for any Borel } B \subseteq \mathcal{Y} \times \mathcal{Y}.$$

We claim $\tilde{\gamma} \in \Pi(f_{\#}\mu, f_{\#}\nu)$. Indeed, for any Borel set $C \subseteq \mathcal{Y}$, the first marginal of $\tilde{\gamma}$ satisfies

$$\tilde{\gamma}(C \times \mathcal{Y}) = \gamma\big(f^{-1}(C) \times \mathcal{X}\big) = \mu\big(f^{-1}(C)\big) = (f_{\#}\mu)(C),$$

and similarly the second marginal equals $f_{\#}\nu$.

Now compute the transport cost under $\tilde{\gamma}$:

$$\int_{\mathcal{Y} \times \mathcal{Y}} \|u - v\|^2 \, d\tilde{\gamma}(u, v) = \int_{\mathcal{X} \times \mathcal{X}} \|f(x) - f(y)\|^2 \, d\gamma(x, y).$$

Since $f$ is $L$-Lipschitz, $\|f(x) - f(y)\| \le L\|x - y\|$ for all $x, y$, hence

$$\int \|f(x) - f(y)\|^2 \, d\gamma(x, y) \le L^2 \int \|x - y\|^2 \, d\gamma(x, y).$$

Because $W_2^2(f_{\#}\mu, f_{\#}\nu)$ is the *infimum* of the left-hand side over all couplings in $\Pi(f_{\#}\mu, f_{\#}\nu)$, in particular

$$W_2^2(f_{\#}\mu, f_{\#}\nu) \le \int_{\mathcal{Y} \times \mathcal{Y}} \|u - v\|^2 \, d\tilde{\gamma}(u, v) \le L^2 \int_{\mathcal{X} \times \mathcal{X}} \|x - y\|^2 \, d\gamma(x, y).$$

Finally, taking the infimum over $\gamma \in \Pi(\mu, \nu)$ yields

$$W_2^2(f_{\#}\mu, f_{\#}\nu) \le L^2 W_2^2(\mu, \nu),$$

and taking square roots completes the proof. $\qquad\square$

**Theorem A.3** (Wasserstein Contraction (Theorem 4.1)). *For any two cities $A$ and $B$, let segment embeddings be $\{h_i^{(A)}\}_{i=1}^{N_S^{(A)}}$ and $\{h_j^{(B)}\}_{j=1}^{N_S^{(B)}}$ in $\mathbb{R}^d$, with empirical measures*

$$\mu_{segment}^{(A)} = \frac{1}{N_S^{(A)}} \sum_{i=1}^{N_S^{(A)}} \delta_{h_i^{(A)}}, \qquad \mu_{segment}^{(B)} = \frac{1}{N_S^{(B)}} \sum_{j=1}^{N_S^{(B)}} \delta_{h_j^{(B)}}.$$

*Let region embeddings be computed by the aggregation map $\Phi(\mathbf{H}_S) = \mathbf{A}_{SR}^{\top} \mathbf{H}_S$ (see Eq. (22)), inducing empirical measures $\mu_{region}^{(A)}$ and $\mu_{region}^{(B)}$. Then*

$$W_2\left(\mu_{region}^{(A)}, \mu_{region}^{(B)}\right) \le W_2\left(\mu_{segment}^{(A)}, \mu_{segment}^{(B)}\right).$$

*Proof.* Let $\gamma^{\star} \in \Pi(\mu_{segment}^{(A)}, \mu_{segment}^{(B)})$ be an optimal coupling. Define $\tilde{\gamma} := (\Phi, \Phi)_{\#}\gamma^{\star}$. Then $\tilde{\gamma} \in \Pi(\mu_{region}^{(A)}, \mu_{region}^{(B)})$. Therefore,

$$W_2^2(\mu_{region}^{(A)}, \mu_{region}^{(B)}) \le \int \|\Phi(x) - \Phi(y)\|^2 \, d\gamma^{\star}(x, y).$$

Since $\Phi$ is 1-Lipschitz (Lemma A.1), $\|\Phi(x) - \Phi(y)\| \le \|x - y\|$, hence

$$W_2^2(\mu_{region}^{(A)}, \mu_{region}^{(B)}) \le \int \|x - y\|^2 \, d\gamma^{\star}(x, y) = W_2^2(\mu_{segment}^{(A)}, \mu_{segment}^{(B)}).$$

Taking square roots finishes the proof. $\qquad\square$

*Remark* A.4 (When can equality hold?). The inequality becomes equality only in special cases where (i) the optimal transport plan $\gamma^{\star}$ between $\mu$ and $\nu$ satisfies $\|f(x) - f(y)\| = L\|x - y\|$ for $\gamma^{\star}$-almost all $(x, y)$, and (ii) the pushforward coupling $(f, f)_{\#}\gamma^{\star}$ remains optimal for the pair $(f_{\#}\mu, f_{\#}\nu)$. Such conditions typically require $f$ to act as an isometry (up to scaling) on the support of the optimal transport.

For non-isometric aggregations, such as averaging-based region pooling, the contraction is generally strict whenever the inter-city discrepancy contains substantial within-region variations (i.e., high-frequency components), since these components are attenuated by the aggregation. Nevertheless, strict contraction cannot be guaranteed for all distributions, e.g., in the trivial case $\mu = \nu$.

### A.3. Hierarchical Abstraction as Low-Pass Filtering

We now justify Theorem 4.2 from the perspective of Dirichlet energy on graphs. Let $\mathcal{G}_S = (\mathcal{V}_S, \mathbf{A}_S)$ be the segment graph with (undirected) weighted Laplacian $\mathbf{L}_S$.[2] For a graph signal $\mathbf{z} \in \mathbb{R}^{N_S}$, its Dirichlet energy is defined as

$$\mathcal{E}(\mathbf{z}) = \mathbf{z}^\top \mathbf{L}_S \mathbf{z} = \frac{1}{2} \sum_{(i,j)} w_{ij}(z_i - z_j)^2,$$

which measures the total variation of $\mathbf{z}$ over the graph.

**Hard-partition operator.** We first consider the idealized case of a (near) hard partition induced by $\mathbf{A}_{SR}$, where each segment $i$ is assigned to exactly one region $\pi(i) \in \{1, \ldots, N_R\}$. Let $\mathcal{C}_k = \{i : \pi(i) = k\}$ denote the set of segments in region $k$. Define the region-wise averaging operator $\mathbf{P}$ by

$$(\mathbf{P}\mathbf{z})_i = \frac{1}{|\mathcal{C}_{\pi(i)}|} \sum_{u \in \mathcal{C}_{\pi(i)}} z_u, \qquad i \in \mathcal{V}_S.$$

This operator replaces each segment value by the average value of its region, and corresponds to $\mathbf{P} = \mathbf{A}_{SR}\mathbf{A}_{SR}^\top$ in the limiting case where assignments are sharp.

**Lemma A.5** (Block averaging decreases Dirichlet energy). *For the operator $\mathbf{P}$ defined above and any graph signal $\mathbf{z}$,*

$$(\mathbf{P}\mathbf{z})^\top \mathbf{L}_S (\mathbf{P}\mathbf{z}) \leq \mathbf{z}^\top \mathbf{L}_S \mathbf{z}.$$

*Proof.* We express the Dirichlet energy in edge form:

$$\mathcal{E}(\mathbf{P}\mathbf{z}) = \frac{1}{2} \sum_{(i,j)} w_{ij}\big((\mathbf{P}\mathbf{z})_i - (\mathbf{P}\mathbf{z})_j\big)^2.$$

If $i$ and $j$ belong to the same region, then $(\mathbf{P}\mathbf{z})_i = (\mathbf{P}\mathbf{z})_j$ and the corresponding term is zero, which eliminates all intra-region variations.

If $i \in \mathcal{C}_a$ and $j \in \mathcal{C}_b$ with $a \neq b$, then $(\mathbf{P}\mathbf{z})_i = \bar{z}_a$ and $(\mathbf{P}\mathbf{z})_j = \bar{z}_b$, where $\bar{z}_k$ denotes the mean of $\{z_u : u \in \mathcal{C}_k\}$. By Jensen's inequality, using the convexity of the square function,

$$(\bar{z}_a - \bar{z}_b)^2 = \left( \frac{1}{|\mathcal{C}_a||\mathcal{C}_b|} \sum_{u \in \mathcal{C}_a} \sum_{v \in \mathcal{C}_b} (z_u - z_v) \right)^2 \leq \frac{1}{|\mathcal{C}_a||\mathcal{C}_b|} \sum_{u \in \mathcal{C}_a} \sum_{v \in \mathcal{C}_b} (z_u - z_v)^2.$$

Substituting this bound into the edge sum shows that each inter-region contribution after averaging is bounded by the average of the original squared differences. Summing over all edges yields $\mathcal{E}(\mathbf{P}\mathbf{z}) \leq \mathcal{E}(\mathbf{z})$. $\square$

**Theorem A.6** (Hierarchical Abstraction as Low-Pass Filtering (Theorem 4.2)). *Let $\mathbf{P}$ be the region-wise averaging operator induced by $\mathbf{A}_{SR}$ in the hard-assignment limit. Then for any graph signal $\mathbf{z}$,*

$$(\mathbf{P}\mathbf{z})^\top \mathbf{L}_S (\mathbf{P}\mathbf{z}) \leq \mathbf{z}^\top \mathbf{L}_S \mathbf{z},$$

*which implies that hierarchical abstraction acts as a spectral low-pass filter, suppressing high-frequency components while preserving low-frequency structure.*

*Proof.* The result follows directly from Lemma A.5. The spectral interpretation is immediate, since the Dirichlet energy weights Laplacian eigencomponents proportionally to their eigenvalues. $\square$

*Remark* A.7 (Spectral interpretation). The inequality controls the Dirichlet energy, which admits the spectral decomposition $\mathbf{z}^\top \mathbf{L}_S \mathbf{z} = \sum_k \lambda_k \alpha_k^2$. Since larger eigenvalues correspond to higher graph frequencies, the contraction implies that components associated with large $\lambda_k$ are more strongly penalized. Moreover, for region-wise averaging, signals that are smooth within regions (i.e., low-frequency components) are approximately preserved, while within-region oscillations (high-frequency components) are suppressed. Hence, hierarchical abstraction acts as a spectral low-pass filter.

---

[2]For directed road graphs, we adopt the standard symmetrized Laplacian in spectral analysis.

*Table 2.* Statistics of the three datasets after preprocessing.

| Statistics | Beijing | Chengdu | Xi'an |
|---|---|---|---|
| # road segments | 15,042 | 2,857 | 3,686 |
| # road edges | 47,082 | 8,224 | 7,341 |
| # road types | 17 | 13 | 12 |
| graph diameter | 131 | 71 | 47 |
| # driving records | $\sim$16.0M | $\sim$9.6M | $\sim$6.7M |
| # trajectories | 303k | 224k | 493k |
| average hop number | 48 | 35 | 28 |

*Remark* A.8 (Soft assignment). When $\mathbf{A}_{SR}$ is soft (Eq. (5)), the operator $\mathbf{P} = \mathbf{A}_{SR}\mathbf{A}_{SR}^{\top}$ acts as a soft averaging operator. In the low-temperature regime ($\tau \to 0$), the assignments become increasingly peaked and $\mathbf{P}$ converges to the hard block-averaging operator analyzed above; hence the Dirichlet energy contraction holds exactly in this limit. For finite $\tau$, $\mathbf{P}$ provides a continuous relaxation of hard averaging and retains its smoothing nature, systematically suppressing within-region variations. In addition, the explicit smoothness regularizer $\mathcal{L}_{\mathrm{smooth}}$ (Sec. 4.3) guarantees that the learned low-frequency representations remain spatially smooth, reinforcing the low-pass behavior during training.

*Remark* A.9 (When is the inequality strict?). Equality in Lemma A.5 and Theorem A.6 can only occur when the signal is exactly constant within each region, i.e., when it lies entirely in the subspace spanned by region-indicator functions. In practical road network scenarios, however, segment-level representations exhibit substantial intra-region variations due to local geometry and topology. Consequently, the region-wise averaging operation strictly reduces the Dirichlet energy, and the contraction is typically strict in cross-city settings.

## B. Dataset Description

We utilize three real-world road network and trajectory datasets in our experiments. Table 2 summarizes the dataset statistics after preprocessing. All three road networks exhibit large graph diameters, and the average hop number along trajectories is also considerable, reflecting the long-range dependencies and complex topology of urban-scale transportation systems. Moreover, the three cities present markedly different macroscopic layouts: grid-like (Beijing), ring-structured (Chengdu), and hybrid irregular (Xi'an), which induces substantial cross-city structural heterogeneity.

For all datasets, road network information is extracted from *OpenStreetMap*. We consider only road segments as graph nodes, while other geographic entities (e.g., intersections or off-network points) are excluded. Map matching is performed using the open-source tool *FMM* [3], which aligns sampled GPS points to corresponding road segments, transforming raw GPS traces into time-ordered segment sequences. To reduce redundancy caused by high-frequency sampling, consecutive points mapped to the same road segment are merged, retaining only the entry and exit points for each segment.

The *Beijing Taxi* dataset was collected from over 18,000 taxis operating in Beijing, China, from November 1 to November 30, 2011. Each record is represented as a tuple $\langle tid, te, longitude, latitude, state \rangle$, where $tid$ denotes the taxi identifier, $te$ the timestamp, and $state$ indicates passenger occupancy. The state information is used to segment continuous records into individual trips, where the "*No passengers*" state marks the end of a trip.

The *Chengdu Taxi* and *Xi'an Taxi* datasets are released by the *DiDi GAIA Open Dataset* platform. Each dataset contains one month of complete trajectory data (from November 1 to November 30, 2016) for all DiDi-operated taxis within the second ring roads of Chengdu and Xi'an, respectively. Each individual ride is treated as a separate trajectory. Although the datasets are collected in different years, our focus is on learning structural road representations rather than modeling temporal traffic dynamics, making them suitable for cross-city evaluation.

## C. Baseline Model Details

We provide detailed descriptions of the baseline models and explain how they are adapted to the road network setting. Following the main paper, we group baselines into three categories: random-walk-based models, spatial-based GNNs (including attention-based graph transformers), and spectral-based GNNs.

---

[3] https://www.github.com/cyang-kth/fmm

- **Random Walk-based Models.** These methods learn node embeddings by generating random walk sequences and applying shallow embedding techniques.

  *DeepWalk* (Perozzi et al., 2014) learns embeddings by treating truncated random walks as sentences and optimizing a Skip-Gram objective. We adapt it by performing random walks on the road segment graph.

  *IRN2Vec* (Wang et al., 2019b) is originally designed for intersection representation learning by leveraging geo-locality and mobility patterns. We adapt it by replacing intersections with road segments and using segment-level geographic and type-related attributes.

  *Toast* (Chen et al., 2021) improves random-walk embeddings by introducing an auxiliary traffic context prediction objective. We apply it to segment graphs using the same trajectory preprocessing pipeline as our method.

- **Spatial-based GNN.** These methods aggregate information directly in the spatial domain through message passing, hierarchical pooling, or attention mechanisms.

  *GeomGCN* (Pei et al., 2020) incorporates geometric relationships to improve neighborhood aggregation. We apply it to road segment graphs following the default design for node representation learning.

  *HRNR* (Wu et al., 2020) is a hierarchical road network representation learning framework that performs multi-level pooling guided by semantically informed assignments and trajectory signals. We follow the standard implementation.

  *GT* (Dwivedi & Bresson, 2021) generalizes the Transformer architecture to graphs via self-attention, where Laplacian eigenvectors are used as positional encodings. We apply it to segment graphs without using edge attributes.

  *NFormer* (Wu et al., 2022) introduces scalable attention by approximating softmax attention with kernel functions. We apply it directly to segment graphs following the official implementation.

- **Spectral-based GNNs.** These methods design learnable filters based on graph spectral principles.

  *GCN* (Kipf & Welling, 2017) can be viewed as a first-order approximation of spectral graph convolution. We apply it to segment graphs and train it in a supervised manner for downstream tasks.

  *FAGCN* (Bo et al., 2021) enables adaptive filtering to capture information beyond the low-frequency band. We use it as a representative adaptive spectral baseline.

  *ChebNetII* (He et al., 2022) revisits Chebyshev polynomial approximation for graph convolution and enables flexible high-order spectral filtering. We apply it to segment graphs with the recommended filter order settings.

  *HiFiNet* (Ma et al., 2026) performs hierarchical frequency decomposition for road network representation learning. We follow the original implementation and use it as the strongest spectral baseline.

## D. Evaluation Tasks and Metrics

We evaluate the comparison methods on three traffic-related application tasks and corresponding evaluation metrics. For each task, we construct a simple and standard neural network architecture (*e.g.,*, GRU or MLP) as the basic framework and incorporate the learned road network representations (mainly for road segments) as input embeddings. We intentionally avoid complex architectures or auxiliary data, and use the same downstream architectures and training protocols for all methods to ensure fair comparison. The evaluation tasks and metrics are described as follows:

• *Next Location Prediction:* This task aims to predict the next location a user will visit (Wu et al., 2017). We construct a GRU-based model that takes historical trajectories as input and outputs a ranked list of candidate road segments. To emphasize long-range dependencies, trajectories are down-sampled at ten-minute intervals. Performance is evaluated by how highly the ground-truth next location is ranked in the candidate list (ACC@1 and ACC@5).

• *Label Classification:* This is a standard task for evaluating representation learning models (Wang et al., 2019b). Our dataset provides semantic labels for road segments (*e.g.,*, *bridge* and *tunnel*). We use a lightweight MLP classifier that takes segment representations as input and outputs label distributions. Performance is measured using F1-score and AUC.

• *Destination Prediction:* This task aims to predict the final destination given a partial (prefix) trajectory (Xue et al., 2013). We construct a GRU-based predictor that takes segment representations as input and predicts the destination, defined as the last location in the trajectory. Performance is evaluated using ACC@1 and ACC@5.

# E. Computational Complexity

We analyze the computational complexity of CoSpec for a single forward pass on one city. Let $N_S$ and $E_S$ denote the number of road segments and edges in the segment graph, and $N_R$ denote the number of regions (or prototypes), with $N_R \ll N_S$. The embedding dimension is $d$, and the Chebyshev orders for the low- and high-frequency paths are $K_L$ and $K_H$, respectively.

**Segment-level encoding.** The initial feature embedding and shallow GNN encoder operate on the segment graph, incurring $\mathcal{O}(N_S d + E_S d)$ time, which is linear in the graph size.

**Region assignment and interaction.** The soft assignment between segments and regions is implemented via cross-attention, whose dominant cost comes from computing segment–region similarities, leading to $\mathcal{O}(N_S N_R d)$ complexity. Region feature aggregation and up-/down-projection share the same order. The subsequent Region Graph Transformer introduces a quadratic term $\mathcal{O}(N_R^2 d)$, which is moderate in practice since $N_R \ll N_S$.

**Dual-path spectral propagation.** Both the low-frequency and high-frequency paths rely on Chebyshev graph convolutions on the segment graph. Their combined complexity is $\mathcal{O}((K_L + K_H)E_S d)$, where $K_L$ and $K_H$ are small constants. Other operations, such as FiLM modulation, spectral fusion, and smoothness regularization, are linear in $N_S$ and do not affect the leading order.

**Overall complexity.** Combining the dominant terms, the total time complexity of CoSpec is

$$\mathcal{O}\big(N_S N_R d + (K_L + K_H)E_S d + N_R^2 d\big). \tag{25}$$

In typical settings, $N_R \ll N_S$ and $K_L, K_H$ are small, making the model near-linear in the segment graph size, with the segment–region assignment as the main additional overhead.

**Memory complexity.** The memory cost is dominated by storing segment and region embeddings, $\mathcal{O}(N_S d + N_R d)$. If the segment–region assignment matrix is stored densely, it requires $\mathcal{O}(N_S N_R)$ memory; in practice, sparse or top-$k$ assignments can be used to reduce this cost.

# F. Implementation Details

All experiments are conducted on a single NVIDIA GeForce RTX 3090 GPU (24GB VRAM) under Ubuntu 20.04 with Python 3.10. Our model is implemented in PyTorch 2.2.0 and trained using the AdamW optimizer. The embedding dimension of road segments is set to $d = 608$, and the number of regions in the hierarchical abstraction is fixed to $N_R = 200$ unless otherwise specified. For spectral modeling, the Chebyshev polynomial orders are set to $K_L = 4$ and $K_H = 2$ for the low- and high-frequency paths, respectively. The region-level Transformer consists of 2 layers with 8 attention heads, using a hidden dimension equal to $d$. City-adaptive prototype modulation is implemented with a low-rank bottleneck of rank $r = 8$. All learnable parameters, including city embeddings and global prototype embeddings, are initialized from a zero-mean Gaussian distribution with a standard deviation of 0.01. The coefficients of all four auxiliary loss terms are set to 0.25. Sparse matrix operations are employed for graph-related computations to improve efficiency, and dropout is disabled during inference while keeping the learned hierarchical assignments fixed.

# G. Further Analysis

### G.1. Feature Correlation Analysis

To rigorously quantify the disentanglement quality and validate cross-city feature alignment, we perform a feature correlation analysis. For each city, we extract the learned low-frequency and high-frequency representations. We then compute the Pearson correlation matrix between all feature dimensions. To eliminate city-specific biases and capture domain-invariant model behavior, we report the element-wise average of the absolute correlation matrices across all three datasets (Beijing, Chengdu, Xi'an).

The resulting visualization (Figure 6) divides the matrix into diagonal blocks (intra-band consistency) and off-diagonal blocks (inter-band leakage). Although both models achieve approximate orthogonality (near-zero cross-correlations), a

**Feature Disentanglement Analysis: Orthogonality Check**

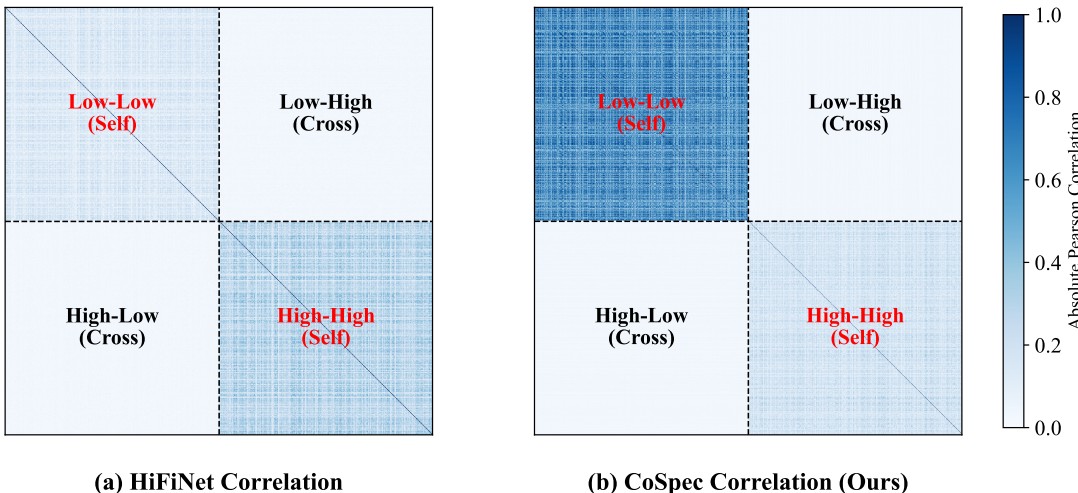

(a) HiFiNet Correlation        (b) CoSpec Correlation (Ours)

*Figure 6.* **Feature Correlation Analysis.** Visualization of Pearson correlation matrices averaged across cities. **(a) HiFiNet** exhibits weak internal consistency (faint Low-Low) and high redundancy (dark High-High), indicating that it captures city-specific biases rather than shared knowledge. **(b) CoSpec** achieves a dual advantage: high internal consistency in the common subspace (dark Low-Low) and high independence in the specific subspace (whitened High-High), confirming effective semantic alignment and disentanglement.

deeper inspection reveals fundamental disparities driven by their learning objectives:

**(1) Common Knowledge Consistency (Low-Low): HiFiNet** (Fig. 6a) displays weak internal signals (subdued intensity). This indicates that without cross-city constraints, HiFiNet learns *city-specific* low-frequency biases (e.g., local ring topologies) that do not align across domains, resulting in signal washout upon averaging. In sharp contrast, **CoSpec** (Fig. 6b) exhibits intense internal correlations (dark blue). This demonstrates *emergent semantic alignment*: our contrastive objective acts as a filter, forcing the model to discard city-specific variations and converge to a compact, domain-invariant manifold for common knowledge.

**(2) Specific Knowledge Independence (High-High): HiFiNet** exhibits unexpectedly high correlations in the high-frequency band. Driven by a pure reconstruction objective, the model implicitly leaks global structural patterns into the specific band to minimize error, leading to substantial feature redundancy. Conversely, **CoSpec** maintains low correlations (whitened block), verifying that the "alignment pressure" on the low-frequency stream successfully strips common structures out of the specific band. This ensures that the high-frequency components encode only diverse and independent local fingerprints, achieving robust disentanglement.

### G.2. Prototype-Level Visualization

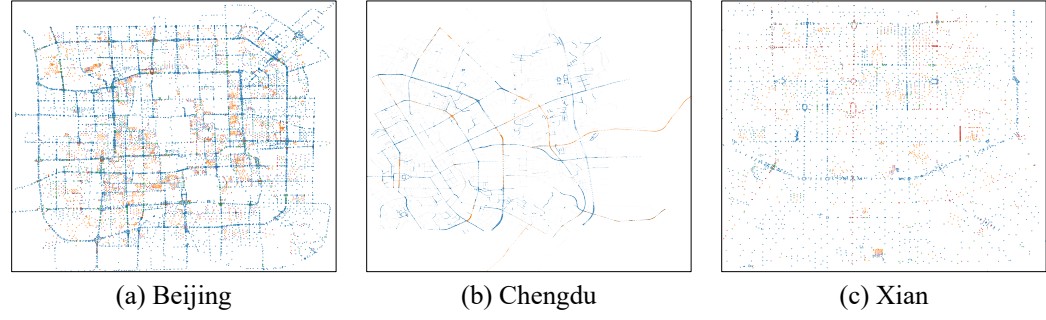

(a) Beijing        (b) Chengdu        (c) Xian

*Figure 7.* Prototype-level visualization of road segments in (a) Beijing, (b) Chengdu, and (c) Xi'an. Each point corresponds to a road segment, and different colors indicate learned functional prototypes. Major arterial roads in different cities are predominantly assigned to the same prototype, suggesting transferable low-frequency functional semantics.

Figure 7 visualizes the learned prototype assignments of road segments in Beijing, Chengdu, and Xi'an. Each point corresponds to a road segment, and different colors indicate the functional prototype to which the segment is assigned. The visualization is produced using the learned region-level prototypes, without using any city-specific labels.

Despite the substantial structural differences among the three cities, we observe a consistent pattern across all cases: major arterial roads are predominantly assigned to the same prototype (shown in blue). These roads typically form the backbone of urban transportation systems and correspond to large-scale, functionally similar components across cities.

This phenomenon provides qualitative evidence that the learned prototypes capture transferable low-frequency functional semantics. While local streets and secondary roads exhibit more diverse prototype assignments, reflecting city-specific geometric variations, the alignment of primary road structures suggests that our method successfully separates shared low-frequency commonalities from high-frequency specificities. Such prototype-level consistency supports the effectiveness of region-based abstraction for cross-city representation alignment.

