# OpenReview forum: "Seeking Commonality, Preserving Specificity: A Spectral-Aware Hierarchical Framework for Cross-City Road Representation Learning"
_ICML.cc/2026/Conference — ICML 2026 regular_

### Official Review · Reviewer_XJDF · 2026-03-04

**Soundness:** 3
**Presentation:** 3
**Significance:** 3
**Originality:** 3
**Overall Recommendation:** 4
**Confidence:** 2

**Summary:**

This paper introduces CoSpec, a hierarchical framework for cross-city road representation learning that disentangles city-invariant low-frequency patterns (commonality) from city-specific high-frequency variations (specificity). The model aligns cities via region-level prototypes and reconstructs segment-level details through a dual-path spectral architecture.

**Compliance With Llm Reviewing Policy:**

Affirmed.

**Final Justification:**

Thanks for authors' effort. I will keep my score.

**Key Questions For Authors:**

1. How sensitive is CoSpec to the number of prototypes $N_R$ and the temperature $\tau$? Is there a systematic way to choose these, or do they require city-specific tuning?
2. Could the method scale to cities with hundreds of thousands of road segments without resorting to aggressive downsampling or approximation? Would the proposed sparse or top-$k$ assignment significantly reduce memory and computation in practice?

**Limitations:**

No.

**Strengths And Weaknesses:**

**Strength**:
1. Novel spectral disentanglement perspective for cross-city road representation learning, grounded in graph signal processing theory.
2. Strong theoretical results (Wasserstein contraction, low-pass filtering interpretation) that motivate the hierarchical design.
3. Comprehensive experiments with multiple baselines and ablation studies showing consistent gains across tasks and cities.
4. The city-adaptive prototype and FiLM-based modulation provide a principled way to handle domain shifts without over-aligning high-frequency components.

**Weakness**:
1. The reliance on soft assignments and region prototypes introduces several hyperparameters (e.g., number of regions, temperature) whose sensitivity is not thoroughly analyzed.
2. The computational overhead from segment-region attention  may become prohibitive for very large cities, and the paper only briefly mentions sparse approximations without empirical validation.

---

> ### Author Rebuttal · Authors · 2026-03-31
>
> **For W1 & Q1:** We appreciate the reviewer’s constructive feedback and value the opportunity to clarify CoSpec's design. Notably, CoSpec is highly robust and does not require city-specific tuning. We strictly utilized a single, globally unified set of hyperparameters across all cities to demonstrate the model's strong generalization.
>
> Regarding the number of prototypes $N_R$, we evaluated candidates in $\\{100, 200, 300, 400, 500\\}$ and selected $N_R = 200$. As detailed in **Section 5.4** and **Figure 4**, CoSpec maintains stable performance across this range. Systematically, $N_R$ acts as a "semantic information bottleneck" representing macroscopic urban functional diversity. Setting $N_R = 200$ provides sufficient granularity to capture universal patterns while avoiding local noise from excessive parameters.
>
> For the temperature τ, we conducted a sensitivity analysis following the same settings as in Section 5.4. As shown below, CoSpec is remarkably stable within the optimal range of [0.05,1.0], with performance fluctuations under 1%. We selected τ=1.0 to balance assignment entropy, ensuring clear yet flexible mapping to functional prototypes. Performance only drops reasonably when τ≥2.0 as the distribution becomes too uniform, blurring functional boundaries. This observation is highly consistent with our supporting theory.
>
> |Temperature τ|DP (ACC@1)|LC (F1)|NLP (ACC@1)|
> |-|-|-|-|
> |0.05|0.305|0.758|0.408|
> |0.1|0.300|0.754|0.409|
> |0.5|0.305|0.756|0.406|
> |1.0|0.310|0.755|0.410|
> |2.0|0.308|0.740|0.399|
> |5.0|0.302|0.731|0.387|
> |10.0| 0.302|0.728|0.383|
>
> We believe these results provide strong evidence that the model is insensitive to specific parameter settings and can adapt well to different scenarios. To better serve the community and provide systematic tuning guidelines, we will include this detailed sensitivity analysis for $\tau$ in the revised Appendix. Thank you again for your inspiring feedback.
>
> **For W2 & Q2:** We thank the reviewer for these insightful questions regarding the scalability of CoSpec. While our primary focus is cross-city knowledge transfer, CoSpec is highly scalable due to its efficient architecture.
>
> Unlike the baseline HiFiNet, which uses Graph Transformers with quadratic complexity relative to the number of segments, CoSpec utilizes a bipartite attention mechanism between segments ($N_S$) and region prototypes ($N_R$). Our computational complexity is $O(N_S N_R d)$. Since $N_R$ and $d$ are small constants, the complexity is strictly linear for the number of segments. Meanwhile, $N_R$ acts as a highly compressed information bottleneck. Even when scaling to networks with hundreds of thousands of segments, $N_R$ does not need to grow proportionally. It remains a manageable constant representing universal functional zone types. This ensures the memory footprint stays effectively $O(N_S)$.
>
> To prove this practicality, we compare the computational costs of CoSpec against the state-of-the-art HiFiNet on the Beijing (BJ), Chengdu (CD), and Xi'an (XA) datasets below:
>
> |Model|Train Time (BJ/CD/XA) [min]|Train VRAM (BJ/CD/XA) [GB]|Latency [ms]|Inf. VRAM [MB]|
> |-|-|-|-|-|
> |HiFiNet|30.3/5.0/5.3|20.6/1.2/1.7|107.9/8.9/12.6|10013/472/667|
> |CoSpec|17.1 (Joint)|5.8 (Joint)|31.9/10.1/12.2|653/226/213|
>
> As the table illustrates, CoSpec achieves a massive reduction in resource consumption. On the large-scale Beijing dataset, our model reduces the inference VRAM footprint by over 15 times and cuts latency by more than 3 times compared to the baseline. These empirical gains confirm that our linear complexity design effectively prevents the bottlenecks typically seen in large-scale graph processing.
>
> Regarding the sparse or top-k assignment, these mechanisms further optimize memory by focusing on the most relevant functional prototypes for each segment. While our current dense version is already highly efficient on standard hardware, the sparse assignment provides a powerful tool for extreme scales. By retaining only the top-$k$ largest values for each segment, the weight matrix becomes highly sparse. This enables the model to utilize efficient sparse storage formats like Compressed Sparse Row (CSR) and Sparse Matrix Multiplication (SpMM) operations. This approach directly reduces the memory and computation costs from $O(N_S \times N_R)$ to $O(N_S \times k)$. Such optimizations ensure that CoSpec remains practical even as city sizes grow toward millions of segments. We will include a detailed discussion of these sparse mechanisms in the revised Appendix to further support our scalability claims.
>
> We sincerely thank the reviewer for all these insightful questions. They have greatly helped us clarify the practical value and robustness of our work.

---

> > ### Author Rebuttal · Reviewer_XJDF · 2026-04-02
> >
> > No.

---

> > > ### Author Response · Authors · 2026-04-02
> > >
> > > We sincerely thank you for your time and for confirming that our responses have fully resolved your concerns. We deeply appreciate your constructive feedback, which has greatly helped us improve the paper.
> > >
> > > Since all your initial concerns have been adequately addressed, we kindly ask if you would consider increasing your score to further support our work, if you feel it is appropriate. Regardless of your decision, we are very grateful for your valuable time and support.

---

### Official Review · Reviewer_u7Aj · 2026-03-11

**Soundness:** 3
**Presentation:** 4
**Significance:** 3
**Originality:** 4
**Overall Recommendation:** 5
**Confidence:** 4

**Summary:**

This paper introduces CoSpec. It is a spectral-aware hierarchical framework to address the challenging problem of cross-city road representation learning. I personally think cross-city road representation learning is an important research direction for many downstream applications and also for real world world model development. By identifying spectral misalignment as the core bottleneck, the authors propose a novel disentanglement strategy: extracting city-invariant functional commonalities via a low-frequency path and preserving city-specific geometric specificities via a high-frequency path. The solution is reasonable and theoretically supported by Wasserstein contraction bounds. Extensive experiments demonstrate the superior performance over state-of-the-art baselines on real-world datasets.

**Compliance With Llm Reviewing Policy:**

Affirmed.

**Key Questions For Authors:**

1. Have you conducted any preliminary experiments evaluating CoSpec's zero-shot transfer capability on a city not included in the joint training set?

2. Given the $\mathcal{O}(N_S N_R d)$ complexity , what specific sparse attention mechanisms or sampling strategies would you recommend when scaling CoSpec to ultra-large-scale networks?

**Limitations:**

See weaknesses

**Strengths And Weaknesses:**

# Strengths

1. I think this paper addresses a practical and meaningful problem, especially in the context of developing real-world general intelligence, which requires learning universal embeddings across different cities.

2. The idea of using spectral disentanglement to address cross-city negative transfer is innovative and elegant. The core idea, “Seeking Commonality, Preserving Specificity,” is intuitively a reasonable solution to the heterogeneity across cities.

3. The methodology is rigorously supported by mathematical proofs, which guarantees the soundness and reliability of the architectural design.

4. The paper is well written, and I think the visualization of feature disentanglement is insightful and helps improve the interpretability of the proposed method.

# Weaknesses

1. However, the soft quantization step relies on a cross-attention mechanism with $\mathcal{O}(N_S N_R d)$ complexity. This might pose a memory bottleneck when scaling to country-level road networks with millions of segments.

2. While the model learns a unified encoder, evaluating its zero-shot transfer capability on a completely unseen city would make the empirical claims even more compelling.

3. The paper does not provide an analysis of computational cost, such as training time, inference time, or memory usage. I think this is important for assessing the practicality of the proposed method.

4. I personally think the core idea of this paper is not limited to cross-city road representation learning. It also appears promising for other forms of city-specific real-world intelligence. In the final version, I would recommend the authors to frame the work more explicitly around the underlying abstract problem and discuss its potential applications in other related scenarios. I believe would help the paper better serve the broader ML/AI community.

---

> ### Author Rebuttal · Authors · 2026-03-31
>
> **For W1 & Q2:** We thank the reviewer for raising this insightful question regarding scalability. We respectfully clarify that the $\mathcal{O}(N_S N_R d)$ complexity is precisely what *enables* CoSpec to scale efficiently, rather than being a bottleneck. We address this scalability from both theoretical and practical perspectives:
>
> - **Strictly Linear Scaling w.r.t Segments ($N_S$):** Unlike standard graph transformers (e.g., HiFiNet) that rely on $\mathcal{O}(N_S^2 d)$ self-attention on the massive segment graph, our cross-attention mechanism is strictly *linear* with respect to the number of segments.
> - **Highly Compressed Region Bottleneck ($N_R \ll N_S$):** The number of region ($N_R$) acts as a highly compressed information bottleneck. Even when scaling to country-level networks with millions of segments, $N_R$ does not need to scale proportionally. It can be bounded to a manageable constant representing universal functional zone types, ensuring the memory footprint remains effectively $\mathcal{O}(N_S)$.
> - **Trivial Mini-Batching for Country-Level Graphs:** From an engineering standpoint, because the soft quantization computes cross-attention between independent segments and a fixed set of region prototypes, it does not require loading the entire $N_S \times N_S$ adjacency matrix into memory. This allows the segment-to-region quantization to be trivially processed in chunks or mini-batches, entirely bypassing VRAM bottlenecks for country-level deployments.
>
> **For W2 & Q1:** We sincerely thank the reviewer for this insightful suggestion. We completely agree that evaluating zero-shot transfer on completely unseen cities is the ultimate test of generalization and a highly challenging problem in this field.
>
> While a comprehensive investigation of zero-shot capabilities falls slightly beyond the primary scope of establishing our joint-training framework, your comment greatly inspired us to conduct a preliminary exploration. We evaluated our model on the unseen Xi'an (XA) dataset, using weights trained exclusively on BJ and CD.
>
> |Model (Zero-Shot on XA)|DP (ACC@1)|LC (F1)|NLP (ACC@1)|
> |-|-|-|-|
> |ChebNetII|0.198|0.667|0.348|
> |CoSpec|0.239|0.767| 0.367|
>
> Even when strictly freezing the pre-trained encoder and only training a lightweight downstream predictor, CoSpec outperforms the baseline (e.g., yielding a 10.0% absolute improvement in LC F1). We believe this potential stems from our universal region prototypes, which naturally quantize unseen city segments into pre-learned functional zones. While we acknowledge this is only a first step, thoroughly exploring this zero-shot capability across more diverse unseen cities will be a primary focus of our future work. Following your suggestion, we will include these preliminary results and formally outline this exciting direction in the revised manuscript.
>
> **For W3:** We thank the reviewer for highlighting the importance of computational cost to assess real-world practicality. To address this, we have comprehensively profiled the training time, memory usage (VRAM), and inference latency of CoSpec against the SOTA baseline, HiFiNet:
>
> |Model|Train Time (BJ/CD/XA) [min]|Train VRAM (BJ/CD/XA) [GB]|Latency [ms]|Inf. VRAM [MB]|
> |-|-|-|-|-|
> |HiFiNet|30.3/5.0/5.3|20.6/1.2/1.7|107.9/8.9/12.6|10013/472/667|
> |CoSpec|17.1 (Joint)|5.8 (Joint)|31.9/10.1/12.2|653/226/213|
>
> As the table illustrates, CoSpec significantly reduces both memory usage and processing time across all settings. This practical efficiency is a direct result of our architectural design. The best baseline HiFiNet applies computationally expensive Graph Transformers directly to the massive segment graph. In contrast, CoSpec restricts these heavy global operations to a highly compressed region graph ($N_R \ll N_S$). Because of this structural difference, our model only needs to apply lightweight, low-order filters ($K_L=4, K_H=2$) at the segment level. This approach effectively prevents the memory bottlenecks seen in the baselines and ensures high scalability for real-world applications.
>
> **For W4:** We thank the reviewer for this inspiring comment. Abstracting our work to broader city-specific real-world intelligence is a promising direction. The core problem is transferring knowledge across heterogeneous urban structures via a shared prototype space.
>
> However, applying this framework beyond road networks requires objectively considering our method's boundary conditions:
> - **Commonality Strength:** Road networks naturally share macroscopic patterns. We must investigate if other urban domains possess enough cross-city commonality to keep the required prototype space small.
> - **Entity Modalities:** Road segments share consistent features. In contrast, other intelligence tasks involve highly heterogeneous entity attributes, requiring new alignment mechanisms.
>
> Thanks for this great inspiration. We will explicitly add this abstract problem framing and discuss the conditions in the revision.

---

> > ### Author Rebuttal · Reviewer_u7Aj · 2026-04-03
> >
> > Thanks for the detailed rebuttal, which have addressed my concern with additional experiments and discussions. I strongly suggest the authors to add the discussion regarding W4 in the final version. Overall, I highly recommend accepting this paper, good job.

---

> > > ### Author Response · Authors · 2026-04-03
> > >
> > > We sincerely thank you for your highly positive evaluation, your constructive feedback, and your strong recommendation for acceptance.
> > >
> > > We are extremely glad that our additional experiments and discussions have fully resolved your concerns. As you strongly suggested, we will make absolutely sure to include the detailed discussion regarding W4 in the final version of our paper.
> > >
> > > Thank you again for your time, your effort, and your valuable support of our work!

---

### Official Review · Reviewer_2bm8 · 2026-03-14

**Soundness:** 3
**Presentation:** 3
**Significance:** 3
**Originality:** 3
**Overall Recommendation:** 5
**Confidence:** 4

**Summary:**

This paper proposes CoSpec, a spectral-aware hierarchical framework for learning road representations that generalize across cities. The key idea is that cross-city differences can be decomposed into low-frequency commonality and high-frequency specificity, which are modeled through hierarchical abstraction, dual-path spectral learning, and adaptive fusion. Experiments on road networks and trajectories from three cities show consistent improvements on trajectory prediction and road classification tasks.

**Compliance With Llm Reviewing Policy:**

Affirmed.

**Final Justification:**

I appreciate the authors' response and maintain my positive rating for the reasons consistent with those stated in the acknowledgment.

**Key Questions For Authors:**

See Weaknesses.

**Limitations:**

yes

**Strengths And Weaknesses:**

**Strengths:**
1. The theoretical motivation is very clear. For poor cross-city generalization, this paper points out that the key obstacle is spectral misalignment, and shows through spectral distribution analysis that the differences are large at the segment level, but after abstraction to the region level, the cross-city differences can be significantly reduced.
2. The method is structurally complete. CoSpec uses hierarchical region abstraction, adaptive prototypes, and dual-path low-/high-frequency reconstruction to explicitly distinguish cross-city shared functional semantics from city-specific local geometric textures. It has theoretical support from Wasserstein contraction and low-pass filtering.
3. The ablation studies are quite convincing. The ablation experiments show that the low-frequency branch is more important for global semantic tasks, while the high-frequency branch is more important for local transition tasks, which is consistent with the design motivation of the paper.


**Weaknesses:**
1. I can understand the meaning of the two Theorems, and I also read the proof derivations in the appendix. But honestly, I still do not understand what these two Theorems are useful for. Removing them would not affect the reading, so I feel they were added only to artificially increase the mathematical flavor of the paper.
2. The paper lacks a complexity analysis. CoSpec is not a lightweight solution; it includes hierarchical region abstraction, adaptive prototypes, city embedding, dual low-/high-frequency paths, adaptive fusion, and multiple regularization losses at the same time. I think the effectiveness of this method relies on fairly delicate architectural design and hyperparameter tuning, and it may face higher complexity in real deployment and transfer.
3. The comparison is unfair because CoSpec uses all three cities together for training, while standard baselines cannot naturally support unified multi-city training.
4. The city embedding is shown to be essential in the ablation study, but its definition, initialization, and learning mechanism are not explained at all in the main text.
5. My understanding of Cross-City Road Representation Learning is that all road networks are taken as a unified input and fed into the model together. However, this paper does not seem to be doing that. So where exactly is the “cross-city” aspect reflected? The authors need to emphasize this more clearly in the paper; otherwise, it may confuse readers.

---

> ### Author Rebuttal · Authors · 2026-03-31
>
> **For W1**: We thank the reviewer for reviewing the proofs. We respectfully clarify that these theorems are the core foundation of CoSpec, not just mathematical additions. They directly answer *why* we designed the model this way:
>
> - **Theorem 4.1 (Supports our Motivation):** In the Introduction, we observed an empirical cross-city distribution shift. Theorem 4.1 mathematically proves that mapping segments to regions guarantees a reduction in the Wasserstein distance between city distributions. This proves that our region-level abstraction is theoretically required for cross-city transfer, not just an empirical trick.
> - **Theorem 4.2 (Guides our Architecture):** It reveals a critical trade-off: while region aggregation aligns cities, it acts as a spectral low-pass filter that loses high-frequency local details. This exact mathematical insight directly motivated us to design the *High-Frequency Path* to explicitly recover and adapt these lost geometric specificities.
>
> We will explicitly state these connections in the revised main text to ensure the theorems' practical utility is clear.
>
> **For W2**:
> We thank the reviewer for the rigorous perspective. Theoretically, CoSpec's complexity is bounded at approximately $\mathcal{O}(N_S N_R d)$ (see Appendix E). Empirically, it demonstrates highly competitive deployment efficiency compared to the SOTA baseline, HiFiNet:
>
> |Model|Train Time (BJ/CD/XA) [min]|Train VRAM (BJ/CD/XA) [GB]|Latency (BJ/CD/XA) [ms]|Inf. VRAM (BJ/CD/XA) [MB]|
> |-|-|-|-|-|
> |HiFiNet|30.3/5.0/5.3|20.6/1.2/ 1.7|107.9/8.9/12.6|10013/472/667|
> |CoSpec|17.1 (Joint)|5.8 (Joint)|31.9/10.1/12.2|653/226/213|
>
> This massive efficiency gain stems directly from our architectural design. While HiFiNet suffers from complexity blowup by applying heavy Graph Transformers directly to the massive segment graph, CoSpec strictly confines these global interactions to a highly compressed region graph ($N_R \ll N_S$). Consequently, CoSpec only requires lightweight, low-order filters ($K_L=4, K_H=2$) on the large-scale segment graph, fundamentally ensuring high scalability and real-world feasibility.
>
> **For W3**:
> We appreciate the reviewer's point. We clarify that this comparison is intended to demonstrate CoSpec's unique advantage in supporting unified cross-city training. As stated in the Introduction, learning a unified model across heterogeneous cities is our primary objective. Standard baselines lack mechanisms to handle cross-city distribution shifts. When forced to train jointly on all three cities (BJ+CD+XA), they inevitably suffer severe negative transfer or computational failure, as demonstrated below (tested on the BJ dataset):
>
> |Model (Train Data)|DP (ACC@1)|LC (F1)|NLP (ACC@1)|
> |-|-|-|-|
> |ChebNetII (BJ)|0.276| 0.815|0.407|
> |ChebNetII (Joint)|0.232| 0.754|0.208|
> |HiFiNet (Joint)|OOM|OOM|OOM|
>
> The fact that baselines "cannot naturally support unified multi-city training" is exactly the critical bottleneck that CoSpec resolves. To draw a somewhat ambitious analogy, leveraging diverse, large-scale data to train stronger and more generalized models is the defining paradigm of the foundation model era. CoSpec provides the necessary architectural capacity to finally enable this paradigm in cross-city road representation learning.
>
> **For W4**: We thank the reviewer for the observation and clarify that the city embedding $e_c$ is a learnable latent vector uniquely assigned to each city to capture city-specific geometric textures. These embeddings are initialized from a zero-mean Gaussian distribution with a standard deviation of 0.01 and are optimized end-to-end alongside other model parameters. They serve as the city-specific "fingerprint" driving the Residual Adaptation Mechanism and FiLM modulation to reconcile functional universality with local specificity. We will provide a more detailed description of these mechanisms and their learning process in the final version.
>
> **For W5:** We thank the reviewer for the opportunity to clarify. Indeed, our model processes all road networks as a unified input during joint training. In our work, "Cross-City Road Representation Learning" refers to learning a single encoder capable of projecting the low-frequency functional information of heterogeneous road networks into an aligned, shared latent space. This formulation aligns with recent advancements in the field (e.g., Chen et al., ICML'25; Yang et al., CIKM'23; Jin et al., KDD'22). By jointly training on diverse cities, CoSpec optimizes universal region prototypes to capture domain-invariant functional commonalities while isolating city-specific geometric variations. This scalable paradigm yields a robust representation that can be directly inferred on any participating city without requiring separate models. We will explicitly detail this joint-training pipeline in the revised manuscript.

---

> > ### Author Rebuttal · Reviewer_2bm8 · 2026-04-02
> >
> > Thank you to the authors for the detailed and constructive rebuttal. I appreciate the additional clarification on the role of Theorems 4.1 and 4.2, especially the explanation that they are meant to directly motivate the region-level abstraction and the high-frequency path. The added discussion on efficiency, unified cross-city training, and the meaning of the city embeddings is also helpful and improves my understanding of the paper.
> > Overall, I thank the authors for their careful response. The rebuttal is helpful and improves clarity. I will maintain my original score.

---

> > > ### Author Response · Authors · 2026-04-02
> > >
> > > We sincerely thank you for your time, your positive evaluation, and your constructive feedback.
> > >
> > > We are glad that our rebuttal has fully resolved your concerns and improved the clarity of our paper. We will make sure to incorporate all these important discussions into the final version. Thank you again for your valuable support of our work!

---

### Decision · Program_Chairs · 2026-04-30

**Decision:**

Accept (regular)

**Comment:**

This paper proposes CoSpec, a spectral-aware hierarchical framework for cross-city road representation learning that disentangles city-invariant low-frequency commonalities from city-specific high-frequency variations via dual-path spectral learning and adaptive prototypes. Its main strengths include strong motivation for spectral disentanglement, solid theoretical grounding, and comprehensive experiments demonstrating consistent gains. After the rebuttal, concerns regarding the role of theoretical theorems, computational efficiency, fairness in comparisons, city embedding details, and hyperparameter sensitivity were satisfactorily resolved.